# NEUROD1 reinforces endocrine cell fate acquisition in pancreatic development

Romana Bohuslavova [1,4], Valeria Fabriciova[1,4], Ondrej Smolik [1,4], Laura Lebrón-Mora [1], Pavel Abaffy [2], Sarka Benesova [2], Daniel Zucha [2], Lukas Valihrach [2], Zuzana Berkova [3], Frantisek Saudek[3] & Gabriela Pavlinkova [1]✉

NEUROD1 is a transcription factor that helps maintain a mature phenotype of pancreatic β cells. Disruption of *Neurod1* during pancreatic development causes severe neonatal diabetes; however, the exact role of NEUROD1 in the differentiation programs of endocrine cells is unknown. Here, we report a crucial role of the NEUROD1 regulatory network in endocrine lineage commitment and differentiation. Mechanistically, transcriptome and chromatin landscape analyses demonstrate that *Neurod1* inactivation triggers a down-regulation of endocrine differentiation transcription factors and upregulation of non-endocrine genes within the *Neurod1*-deficient endocrine cell population, disturbing endocrine identity acquisition. *Neurod1* deficiency altered the H3K27me3 histone modification pattern in promoter regions of differentially expressed genes, which resulted in gene regulatory network changes in the differentiation pathway of endocrine cells, compromising endocrine cell potential, differentiation, and functional properties.

The endocrine pancreas is a key regulatory tissue of glucose metabolism[1]. It consists of the islets of Langerhans, small scarce spherical cell clusters containing five hormone-producing cell types [α cells (glucagon), β cells (insulin), γ cells (pancreatic polypeptide), δ cells (somatostatin) and ε cells (ghrelin)]. In mice, the β-cells are predominately located in the central core of islets, while other hormone-producing cells localized in the periphery, forming a mantle. Dysfunction of these endocrine cells can lead to severe pathophysiology, especially in the case of β-cells failing to produce insulin, resulting in diabetes mellitus. The incidence of diabetes is gradually increasing worldwide, necessitating the development of novel therapies to either compensate for decreased insulin levels or replace dysfunctional β-cells in situ. As the endocrine pancreas bears no signs of postnatal regeneration[2], the solution possibly lies in direct cell reprogramming[3-6] via gene regulatory networks[7,8] driving pancreatic development from multipotent progenitors (MPs) to differentiated endocrine cells[9].

Pancreatic organogenesis starts around embryonic day 8.5 (E8.5) in mice and between 29–33 days post-conception in humans, from MPs expressing a characteristic combination of *Pdx1*, *Sox9,* and *Ptf1a* (reviewed in[9]). As morphogenesis of the pancreas progresses, *Ptf1a* expression becomes limited to acini located at the tips[10], whereas *Pdx1*, *Sox9*, and *Nkx6.1* define bipotent precursors in the trunk area, establishing both ductal and endocrine progenitors (EPs)[11]. Sox9 activates the pro-endocrine gene Neurogenin 3 (*Neurog3*), and thus, initiates endocrine differentiation[12]. Endocrine differentiation in mice proceeds in two temporal waves[13], termed the primary transition starting at E8.5 and secondary transition from E12.5 to E16.5[14], (humans have one wave only[15]). The first differentiated endocrine cells are glucagon⁺ cells, detected in the dorsal pancreas at E9.5. The first insulin⁺ cells, expressing high levels of PDX1, do not appear until E12.5, but their numbers then increase exponentially beginning at E14.5[16]. The upregulation of *Neurog3* is necessary for the induction of cascades of transcription factors orchestrating endocrine cell differentiation,

---

[1]Laboratory of Molecular Pathogenetics, Institute of Biotechnology CAS, 25250 Vestec, Czechia. [2]Laboratory of Gene Expression, Institute of Biotechnology CAS, 25250 Vestec, Czechia. [3]Diabetes Centre, Experimental Medicine Centre, Institute for Clinical and Experimental Medicine, 14021 Prague, Czechia. [4]These authors contributed equally: Romana Bohuslavova, Valeria Fabriciova, Ondrej Smolik. ✉e-mail: gpavlinkova@ibt.cas.cz

such as NEUROD1, INSM1, RFX6, IRX1, and PAX4[9,17,18]. Many involved transcription factors cooperate with epigenetic regulators[19] and together, they direct cell-fate determination. Polycomb group protein complexes play a significant role in this process[20], especially the histone demethylase KDM6B[21]. KDM6B allows the removal of transcriptionally repressive histone modifications from bivalent loci in the genome. Bivalency can be defined as the presence of both activation (H3K4me3) and repression (H3K27me3) marks allowing time-effective changes in the gene expression, and such phenomenon is vital in embryonic development[22–24].

In both mice and humans, *Neurod1* expression follows *Neurog3* directly[25–28], indicating a role of NEUROD1 in the gene regulatory network of the basal endocrine lineage commitment. NEUROD1 serves as an early and comprehensive marker for all endocrine cells[29]. NEUROD1+ EPs leave their proliferative state[30] and begin the process of differentiation into specific endocrine cell types. This change corresponds to the *Neurog3*-low to *Neurog3*-high transition[31]. During this 12h-long cellular process[31,32], a KDM6B-driven switch from bivalency to a H3K4me3-only state occurs in hundreds of promoters, including *Neurod1*, *Insm1,* and *Rfx6*, resulting in gene expression shifting from a *Sox9*, *Onecut1*, and *Hes1*-enriched *Neurog3*-low EP profile towards a *Neurod1*, *Insm1*, *Rfx6*, *Isl1*, and *Pax6* elevated expression pattern leading to endocrine cell differentiation. This transition is accompanied by enhancer priming and activation of genes typical for endocrine functions, predominantly with NEUROD1 binding sites[31], suggesting NEUROD1's pro-endocrine pioneering function.

A pioneering potential of NEUROD1 was reported in neuron differentiation[33]. This pioneering activity involves both activation of cell type-specific genes and repression of genes associated with alternative non-neuronal lineages. NEUROD1 is indispensable for neuronal differentiation, survival, and reprogramming, by reshaping the epigenetic and transcriptional landscape[33–39]. NEUROD1 promotes neuronal fate and preserves neuronal identity by suppressing non-neuronal fates.

Unlike *Neurog3*[40], which exhibits transient expression during endocrine specification, *Neurod1* demonstrates a more dynamic and temporally diverse expression profile across various endocrine cell subtypes throughout development and maturation[41,42]. In mice, *Neurod1* expression initiates around E9.5, persists until E12.5 in early glucagon+ cells, dissipates temporarily, and then peaks again around E17.5 in nascent islets[41]. This spatiotemporal fluctuation is fundamental for the establishment[43] and proper differentiation of the endocrine lineage[44,45]. The significance of NEUROD1 becomes evident when considering the consequences of its deficiency. First, mice lacking *Neurod1* during pancreas development die shortly after birth from severe diabetes partly due to a reduced number of endocrine cells, particularly β cells[44–47]. Second, NEUROD1 is indispensable for β-cell maturation and maintenance[44,48,49]. Third, rare mutations in *Neurod1* in humans manifest as a subtype of maturity-onset diabetes of the young (MODY6)[50–52], since NEUROD1 is one of the direct transcription activators of the insulin gene[53–55], and is therefore, critical for modulating glucose homeostasis. Fourth, *Neurod1* deletion is accompanied by an altered islet architecture[43,45,46,56].

In this study, we focused on molecular cues shaping the endocrine differentiation path associated with *Neurod1*. Using a *Neurod1*Cre endocrine-specific *Neurod1* knockout (*Neurod1*-self terminating; *Neurod1*ST), we introduced an early *Neurod1* deletion disrupting fetal pancreatic development to provide further insight on the role of NEUROD1 in the early endocrine lineage establishment. We identified changes associated with *Neurod1* deletion on a transcriptomic (endocrine specific RNA-seq) and epigenetic level using the CUT&Tag-seq approach, which helps overcome the low-input limits of existing epigenetic-landscape profiling methods such as ChIP-seq[57].

Herein, we link an early endocrine deficiency of NEUROD1 to an altered chromatin landscape of numerous genes resulting in transcriptome dysregulation and ambiguous cell fate commitment, eventually leading to critical developmental defects of the endocrine pancreas, which are manifested both functionally and morphologically.

## Results

### *Neurod1* elimination results in a severe diabetic phenotype

To evaluate the role of *Neurod1* during the early development of the pancreas, we generated a *Neurod1*-"self-terminating" conditional knockout mouse model (*Neurod1*ST) by breeding *Neurod1*loxP/loxP [58] with *Neurod1*Cre/+ [59]. Using CRE-dependent *Ai14*-tdTomato reporter mice[60], we showed that the expression pattern of tdTomato matched the formation of the first differentiated endocrine cell clusters and NEUROD1 expression in the E10.5 pancreas (Supplementary Fig. 1a). We confirmed that *Neurod1*-CRE activity in early endocrine cell clusters emerging from the dorsal pancreatic bud is exclusive to the endocrine lineage in the pancreas and corresponds to the expression of *Neurod1* gene[42,43,61]. The CRE-mediated deletion of *Neurod1* was efficient, with a more than 65% elimination of NEUROD1 in the E10.5 pancreas of *Neurod1*ST (*Neurod1*loxP/loxP; *Neurod1*Cre/+) compared to *Controls* (*Neurod1*loxP/loxP or *Neurod1*loxP/+) (Supplementary Fig. 1b, c).

We observed neonatal mortality in *Neurod1*ST (Supplementary Fig. 2) correlating with a trend of reduced body weight (Fig. 1a) and elevated levels of glycemia (Fig. 1b) measured on the first day after birth (P0). Decreased total pancreatic insulin levels confirmed a severe diabetic phenotype of the *Neurod1*ST (Fig. 1c). No surviving *Neurod1*ST homozygous mutant mice were detected beyond P3. These findings are in line with observations in newborns with global deletion of *Neurod1*[46]. We also characterized the phenotype of mice with the heterozygous genotype (*Het: Neurod1*loxP/+; *Neurod1*Cre/+) containing one functional *Neurod1* allele and *Cre* allele. Mice with *Het* genotype with the respect to development, body weight, blood glucose, and survival were undistinguishable from *Control* mice, confirming no unintended effects of the Cre recombinase and the reduced level of *Neurod1* gene (Fig. 1a, b, and Supplementary Fig. 2). These results allowed us to utilize the Cre driver in combination with the *Ai14*-tdTomato reporter (*Control-Ai14*) for cell-type-specific molecular and morphological analyses.

Next, we evaluated the histology of islets of Langerhans in *Neurod1*ST compared to both *Control* and *Control-Ai14*. We observed reduced endocrine tissue content in the number of glucagon-producing (GCG) α cells and insulin-producing (INS) β cells at P0 (Fig. 1d–f). The percentage of somatostatin (SST)-secreting cells was not affected (Fig. 1g). Consistent with previous studies[44,45], we found a major defect in α- and β-cell proliferation without noticeable increase in apoptotic cells in *Neurod1*ST, as evaluated by TUNEL (Supplementary Fig. 3). The proliferation of β cells in *Neurod1*ST was significantly reduced at both E17.5 (Fig. 1h) and P0 (Fig. 1i). Furthermore, we confirmed previous results[45] indicating no significant effect of *Neurod1* elimination on the α-cell proliferation at E17.5 (Fig. 1h). The β-cell proliferative state in physiological conditions in mice reportedly starts around E17.5 and peaks approximately in P3, preceding a process of maturation[62]. α cells follow the same trend but peak around P0[62]. Concomitantly, α-cell proliferation was moderately reduced in *Neurod1*ST at P0 (Fig. 1i; *p* = 0.0463). Since both β and α cells are a major part of the islet cell mass, a lower proliferation of both β and α cells corresponds to reduced endocrine content of the *Neurod1*ST. Similar to prior reports[44–46], the lack of *Neurod1* was accompanied by disrupted architecture of the islets of Langerhans, forming smaller cell clusters with the distinctive α-cell mantle absent in P0 *Neurod1*ST (Fig. 1i).

To uncover the effects of *Neurod1*-deficiency on the formation, and spatial distribution of the islets of Langerhans in the 3D tissue microenvironment of the pancreas, we performed light sheet fluorescence microscopy at E18.5 on both *Control-Ai14* (Supplementary Movie 1) and *Neurod1*ST-*Ai14* (Supplementary Movie 2), visualizing α-

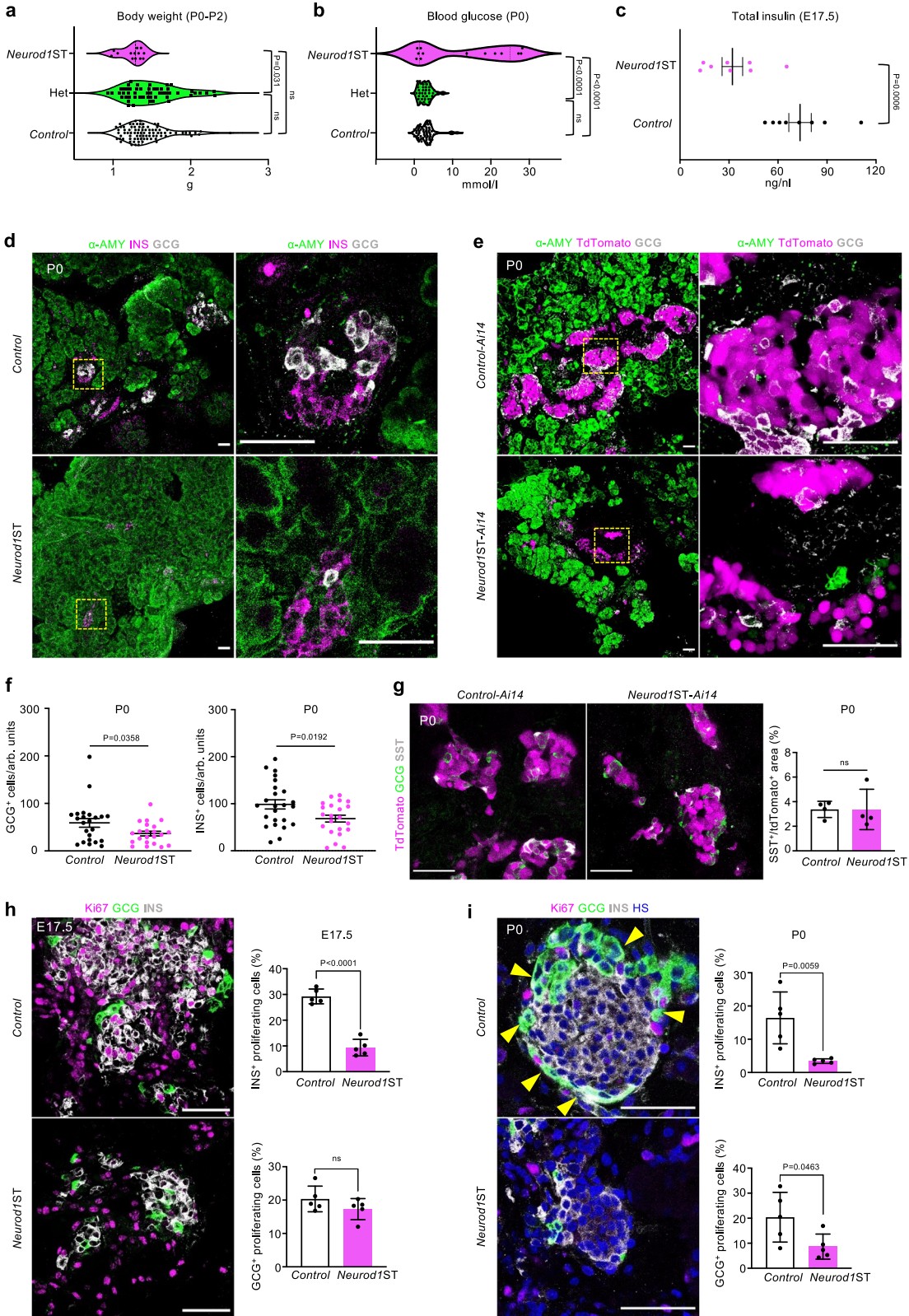

(GLP1; green) and β-cell (INS; white) mass overlap with endocrine tissue (tdTomato; magenta). Surprisingly, apart from a disrupted α-cell islet mantle and increased variability among islet size in *Neurod1*ST, the endocrine cell mass appeared to be semi-organized into strings/sheets along the invisible lines of pancreatic ducts. In contrast, *Control-Ai14* islets shared a more spherical shape, regular size span, and core-mantle organization. Abnormal clustering of islets near ductal

structures may lead to a failure in establishing the normal dispersed islet morphology within the exocrine tissue[63]. This altered organization may disrupt the communication and interactions not only between neighboring islets but also between endocrine and non-endocrine tissues. Consequently, it can disrupt the intricate network that enables intra-islet and inter-islet synchronous hormone secretion[64]. Such alterations in the spatial architecture and communication could have significant

**Fig. 1 | *Neurod1*ST mice demonstrate a severe diabetic phenotype with reduced endocrine cell mass. a** Genotype-determined differences in the measured values of body weight in newborn mice at postnatal day P0-P2 (One-way ANOVA, Tukey's multiple comparisons test). **b** Differences in blood glucose in newborn mice fed *ad libitum* at P0 (One-way ANOVA, Tukey's multiple comparisons test). **c** Total pancreatic insulin levels at embryonic day 17.5, E17.5 (n = 8 pancreases/group). Data are presented as mean ± SEM. **d** Representative immunostaining for α-amylase (α-AMY, a marker of exocrine tissue), glucagon (GCG), and insulin (INS) in pancreatic tissue sections of *Control* and *Neurod1*ST at P0. Boxes indicate the areas magnified in the right panels. **e** Representative microscopy images of tdTomato⁺ cells and immunostaining of α-AMY, and GCG in P0 pancreatic tissue sections of *Control-Ai14* and *Neurod1*ST-*Ai14*. Boxes indicate the areas magnified in the right panels. **f** The number of GCG⁺ and INS⁺ cells counted in the central section with the largest pancreatic footprint of *Neurod1*ST and *Control* pancreases per arbitrary unit (arb.

units = 0.0003 mm³; n = 24 views of area/*Control* from 5 pancreases and 22 views of area/*Neurod1*ST from 5 pancreases). Data are presented as mean ± SEM. **g** Representative images of immunolabeled somatostatin⁺ (SST⁺) cells and quantification of the percentage of SST⁺ per tdTomato⁺ endocrine cells at P0 (n = 4 pancreases/genotype). Data are presented as mean ± SD. **h, i** Quantification of the percentage of proliferating INS⁺ and GCG⁺ cells in the *Neurod1*ST and *Control* pancreas at E17.5 and P0. Representative immunostainings for proliferation marker, Ki67, β-cells expressing INS, and α cells expressing GCG. GCG⁺ cells forming the mantle around INS⁺-cell core in the *Control* are indicated by arrows (in **i**). Note a disrupted islet core-mantle organization in *Neurod1*ST with a missing mantle of GCG⁺ cells around β-cell core compared to *Control*. Nuclei were stained by Hoechst (HS). Data are presented as mean ± SD (n = 5 pancreases/genotype/each age). Unpaired two-tailed *t* test (**c**, **f–i**). Source data for (**a–c**, **f–i**) are provided as a Source Data file. Scale bars, 50 μm. ns not significant.

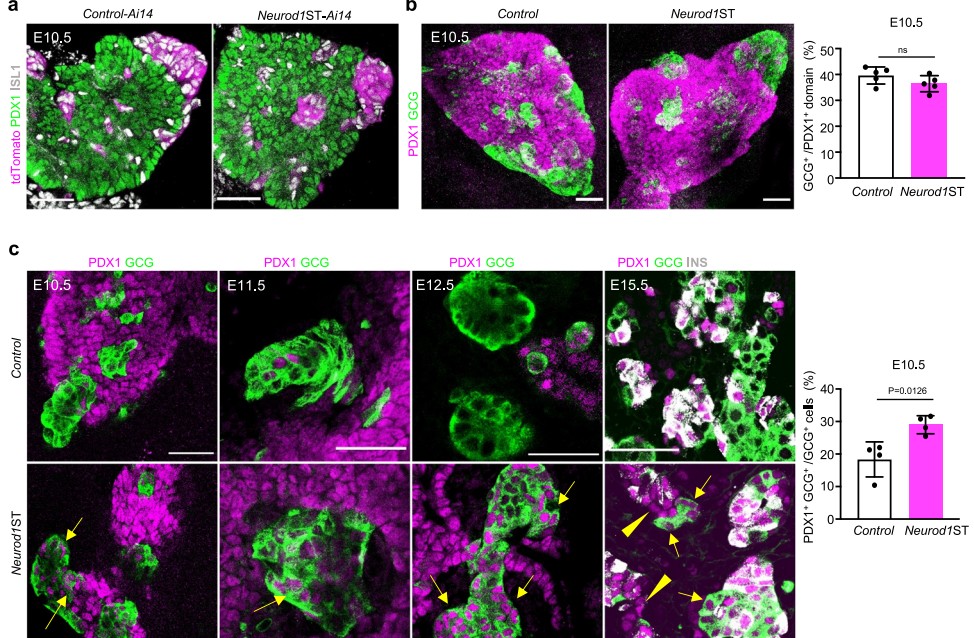

**Fig. 2 | *Neurod1* elimination impairs PDX1 expression in the developing pancreas. a** Representative images of the E10.5 dorsal pancreatic bud during the primary transition visualized by whole-mount immunostaining of PDX1 (a marker of pancreatic progenitors), ISL1 (a marker of pancreatic endocrine cells and pancreas-surrounding mesenchyme), and tdTomato⁺ cell clusters. **b** Whole-mount immunolabeling of glucagon (GCG) clusters in the dorsal pancreatic bud delineated by the expression of PDX1 at E10.5. Semi-quantitative measurements of GCG⁺ area in the PDX1⁺ domain of *Control* and *Neurod1*ST. Data are presented as mean ± SD (n = 5 pancreases/genotype). **c** Representative whole-mount immunostaining images of

the developing pancreas during the primary transition (E10.5, E11.5) and the secondary transition (E12.5, E15.5) show an increased number of cells with PDX1 and GCG co-expression (arrows) in *Neurod1*ST compared to *Control*. Note some PDX1⁺ cells without characteristic insulin (INS) expression within developing islets in *Neurod1*ST at E15.5 (arrowheads). The percentage of cells co-expressing PDX1⁺ and GCG⁺ cells quantified in the *Neurod1*ST and *Control* pancreas at E10.5. Data are presented as mean ± SD (n = 4 pancreases/genotype). Unpaired two-tailed *t* test. Source data are provided as a Source Data file. Scale bars, 50 μm. ns, not significant.

implications for glucose responsiveness and overall endocrine cell function. The 3D endocrine pancreas architecture is a critical factor in maintaining glucose homeostasis and preserving β-cell function, as its disruption has been associated with diabetes mellitus[65,66].

### Endocrine cell differentiation is dysregulated in *Neurod1*ST

The pancreas starts budding from the endodermal foregut around E8.5 and is specified by a multipotent progenitor marker PDX1[67]. Once the endocrine lineage developmentally diverges from its multipotent pro-pancreatic precursors, *Pdx1* expression dissipates and is reconstituted later and only in the β-cell lineage[68,69]. The appearance of the first endocrine cells, GCG-expressing cells, coincides with the onset of *Neurod1* expression during the primary transition in the developing dorsal pancreatic bud at E9.5[43]. However, only a subset of GCG⁺ cells (30%) express NEUROD1 in the E9.5 pancreas[43]. Consistent with the limited requirements of *Neurod1* for the earliest endocrine cells,

efficient reduction of NEUROD1 did not have any significant impact on the formation of endocrine clusters in the dorsal pancreatic bud of *Neurod1*ST at E10.5 (Fig. 2a, b). The relative size of the GCG⁺ cell clusters compared to the PDX1⁺ area of the dorsal pancreatic bud was unchanged. (Fig. 2b). We also observed a persistent expression of PDX1 in the differentiating α-cell lineage in *Neurod1*ST during the primary (E10.5, E11.5) and secondary pancreatic transition (E12.5, E15.5) (Fig. 2c). Although the generation of GCG⁺ endocrine cells was not affected, the persistent expression of PDX1 in GCG⁺ cells, indicates that the differentiation of these cells might be altered or delayed in the absence of NEUROD1.

During the secondary transition of pancreas development marked by a major wave of endocrine cell differentiation, the islets of Langerhans emerge, and the endocrine tissue starts producing all related hormones. A significant larger percentage of the endocrine tdTomato⁺ cells did not produce either GCG or INS hormones at E15.5, suggesting

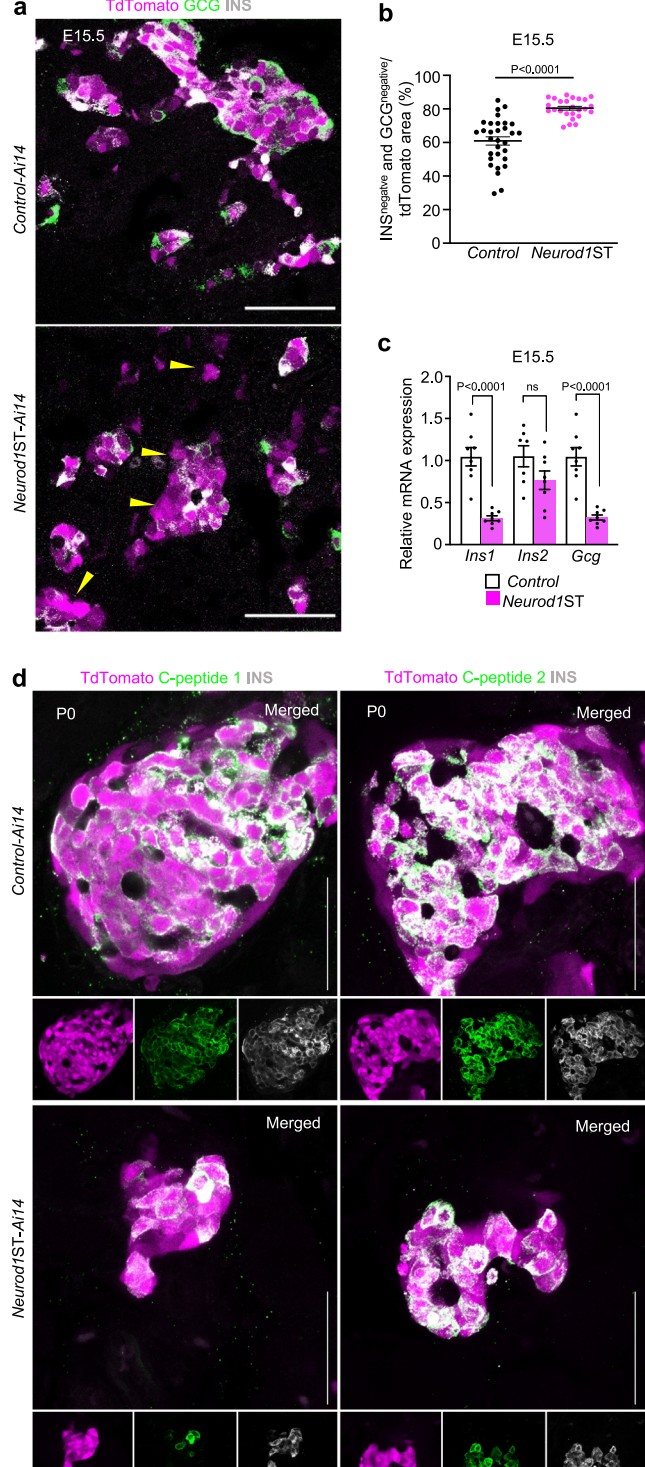

**Fig. 3 | *Neurod1* deficiency results in decreased hormone production.**
**a** Representative confocal microscopy images of tdTomato⁺ cells and immunostaining of glucagon (GCG), and insulin (INS) during the secondary transition at E15.5. Note the increased number of tdTomato⁺ cells without a detectable expression of INS or GCG (arrowheads). **b** Quantification of tdTomato⁺ area without GCG and INS expression in the central section of pancreas with the largest pancreatic footprint. Data are presented as mean ± SEM; (*n* = 31 tdTomato⁺ areas/*Control* from 4 pancreases and 25 tdTomato⁺ areas/*Neurod1*ST from 4 pancreases); unpaired two-tailed *t* test. Source data are provided as a Source Data file. **c** Relative mRNA expression of glucagon (*Gcg*), insulin 1 (*Ins1*), and insulin 2 (*Ins2*) by RT-qPCR using RNA extracted from E15.5 pancreases. Data are presented as mean ± SEM; (*n* = 8 pancreases/genotype); unpaired two-tailed *t* test, Source data are provided as a Source Data file. **d** Representative confocal microscopy images of tdTomato⁺ cells and immunostaining of INS, C-peptide 1 and C-peptide 2 in P0 pancreatic tissue sections. Scale bars, 50 μm. ns, not significant.

data are consistent with previous studies that reported that deletion of *Neurod1* results in a loss of *Ins1* expression, whereas *Ins2* expression is relatively unaffected[45,48]. Therefore, we assessed the expression of peptides C1 and C2, which are cleaved from the corresponding inactive prohormones. At P0, peptide C1 was apparently reduced in *Neurod1*ST-*Ai14* compared to the *Control-Ai14* in contrast to peptide C2 (Fig. 3d), consonantly to different mouse models with conditional *Neurod1* deletions[45,48]. Our results further confirm that loss of *Neurod1* thwarts endocrine lineage differentiation leading to reduced endocrine cell mass and defective hormone production.

## NEUROD1 reinforces endocrine cell fate

To analyze the molecular changes resulting from the elimination of *Neurod1* in developing pancreatic endocrine cells, we performed a bulk-cell-RNA sequencing (bulk-RNA-seq) analysis to evaluate the overall transcriptomic profile in *Neurod1*ST-*Ai14* and validated representative differentially expressed genes using qRT-PCR in independent biological samples (Fig. 4a). Each of four biological replicates per genotype for the bulk-RNA-seq analysis contained 100 tdTomato⁺ FACS-sorted single cells from the dissociated E15.5 embryonic pancreas. tdTomato-expressing cells represent differentiating pancreatic endocrine cells with an active *Neurod1*-promoter.

Differential expression analysis identified 112 downregulated and 153 upregulated protein-coding genes in the *Neurod1*ST-*Ai14* endocrine-specific cell population compared to the corresponding *Control-Ai14* (Fig. 4b, Supplementary Data 1). This analysis confirmed significant reduction of *Ins1* mRNA levels but not *Ins2* in *Neurod1*ST, consistent with the necessity of NEUROD1 for *Ins1* expression[45,48]. Functional enrichment analyses[71] of the set of 112 downregulated genes showed enrichment of Gene Ontology (GO) biological processes associated with endocrine functions such as *regulation of hormone levels, cell-cell signaling, response to monosaccharide, peptide hormone processing, hormone, peptide,* and *insulin secretion* and their *transport* (Fig. 4c, Supplementary Data 1). For example, this analysis identified an enrichment of genes involved in peptide hormone processing, *Cpe*[72], and insulin secretion regulation, *Ffar1*[73], *Nnat*[74], *Scg5*[75], *Pcsk2*[76], *Pcsk6*[77], *Nipal1*[78]. Furthermore, the analysis uncovered the following KEGG pathways[79] and endocrine pathologies: AMPK *signaling, Insulin secretion,* and *Maturity onset diabetes of the young* (MODY), *Type II diabetes mellitus* (T2DM), respectively. *Ins1, Mafa*[80], *Cacna1c*[81], *Irs4*[82] included in T2DM, and MODY, confirm the diabetic phenotype, and *Rapgef4*[83] is involved in the cAMP signaling pathway crucial for Ca²⁺-dependent hormone secretion[84]. Additional genes such as *G6pc2*[85], *Prkag2*[86], and *Hmgcr*[87] are involved in AMP-activated protein kinase (AMPK) signaling, whose activation enhances extracellular glucose uptake[88], and with *Npy*[89] also affects lipid metabolism and cellular physiology[90]. Interestingly, within the Reactome enrichment of downregulated genes, we found genes associated with *Anchoring of the basal body to*

that these tdTomato⁺ cells might have an altered or disrupted differentiation process (Fig. 3a, b). Noticeably, some endocrine cells expressed PDX1 as a marker of differentiating β cells but without INS co-expression in *Neurod1*ST at E15.5 (Fig. 2c). In line with expression dysregulation in differentiating α and β cells, the hormone production at the mRNA level was affected in the *Neurod1*ST pancreas at E15.5 (Fig. 3c). Rodents express two nonallelic insulin genes, *Ins1* and *Ins2*, encoding the proinsulin 1 and proinsulin 2 isoforms[70]. The relative expression of *Ins1* but not *Ins2* mRNA was significantly reduced in *Neurod1*ST compared to *Controls* at E15.5 (Fig. 3c). These

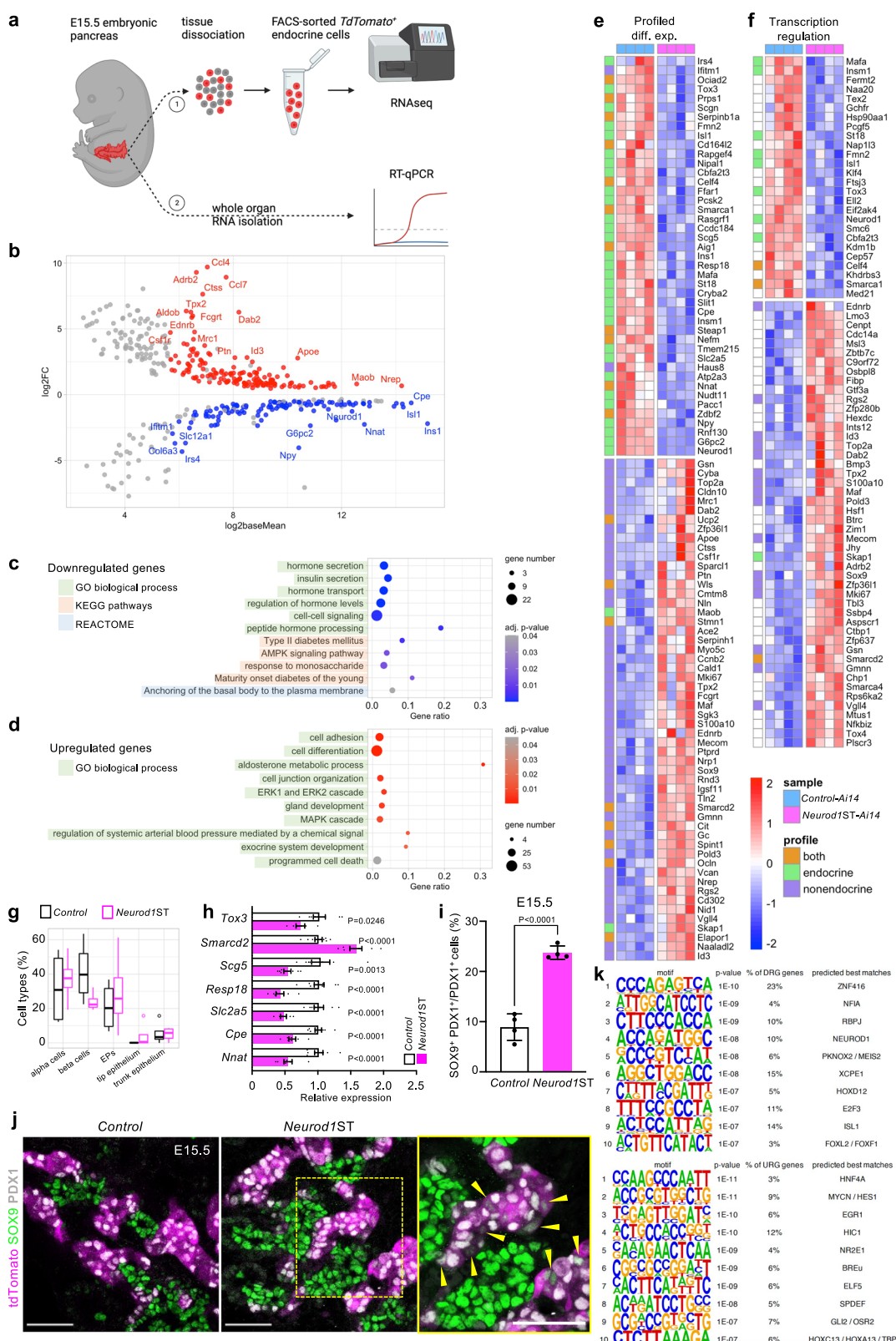

the plasma membrane, which may affect ciliary functions vital for islet development and glucose homeostasis maintenance[91] (Fig. 4c).

The GO analysis of the cohort of 153 upregulated genes showed enrichment of biological processes uncommon in the developing pancreatic endocrine cells (Fig. 4d, Supplementary Data 1), which were related to *exocrine system* and *gland development*, containing characteristic ductal genes, such as multipotency-maintaining *Sox9*[92], the

WNT secretion mediator wntless, *Wls* gene[93,94], or *Ntn4*, a member of the Netrin family, abundantly expressed by pancreatic ductal cells[95]. Thus, the increased expression of non-endocrine genes suggested a flawed differentiation process in *Neurod1*ST. We also found an increased abundance of non-endocrine *cell adhesion* and *cell junction organization* markers such as ductal *Ocln*[96], *Rhoc*[97], *S100a10*[98], *Sparcl1*[99], mesenchymal *Vcan*[100], or acinar *Cldn10*[101], *Pcdh1*[102].

**Fig. 4 | *Neurod1* elimination disrupts gene regulatory network in the developing endocrine cells. a** RNA sequencing experiment design−1: The embryonic pancreases (E15.5) were extracted and dissociated into single cells. Endocrine tdTomato⁺ cells were collected using FACS (100 cells/sample, 4 samples/genotype). Lysed cells total RNA served as template for RNA-seq library preparation. 2: Whole extracted E15.5 pancreases were lysed and used for RT-qPCR validation of several differentially expressed endocrine-specific genes identified by RNA-seq. Created with BioRender.com. **b** RNA-seq data analysis identified 422 differentially expressed genes with 112 downregulated and 153 upregulated protein-coding genes (see also Supplementary data). **c** Downregulated gene functional enrichment (Gene Ontology [GO] biological processes, KEGG pathways, REACTOME). **d** Upregulated GO enrichment (biological processes). **e** Profiled differential expression heatmap: All differentially expressed genes with affiliation to endocrine/non-endocrine single cell expression profile. **f** Transcription regulation heatmap: Differentially expressed genes identified in transcription regulation cluster. **g** Boxplots show the

deconvolved cell type percentage in E15.5 endocrine population from our bulk RNA-seq data (circle marks identified outliers). Centerline of the boxplot indicates the median, the box extends from the 25th to 75th percentiles, and the whiskers represent the rest of the data distribution. **h** RT-qPCR validation of several identified differentially expressed genes at E15.5 in the whole pancreas. Relative mRNA expression data are presented as mean ± SEM ($n = 8$ pancreases/*Control* and 7 pancreases/*Neurod1*ST), unpaired two-tailed $t$ test, ****$p < 0.0001$, ***$p < 0.001$, **$p < 0.01$, *$p < 0.05$. Source data are provided as a Source Data file. **i** Quantification of cells co-expressing SOX9 and PDX1. Data are presented as mean ± SD; ($n = 4$ pancreases/genotype); unpaired two-tailed $t$ test. Source data are provided as a Source Data file. **j** Representative confocal microscopy images of tdTomato⁺ cells and immunolabeled SOX9, insulin (INS), and PDX1 cells. A box indicates the area magnified in the right panel. Arrowheads indicate tdTomato⁺ cells co-expressing SOX9 and PDX1. Scale bars, 50 μm. **k** Top 10 transcription factors identified by de novo motif enrichment analysis of differentially expressed genes.

Interestingly, upregulated *Cdh4* is commonly expressed in both early ductal and endocrine cells but diminishes during embryonic development and remains only in non-endocrine pancreatic tissue[103]. Besides that, we found few upregulated endocrine-related genes, including a membrane raft component *Cd55*[104], β-cell protective *Nid1*[105], and an immune cell adaptor *Skap1*[106]. Notably, 26 genes were linked to *programmed cell death*, which highly overlapped with the cell adhesion cohort. Many of those genes were related to various kinds of neoplasia progression (*Tpx2, Id3, Top2a, Dab2, Mecom, Nrp1*[107–112]), including pancreatic ductal adenocarcinoma, or pancreatitis (*Gsn*[113]). Furthermore, we observed an increased expression in genes involved in the *MAPK/ERK cascade*, which reportedly restrains endocrine specification[114]. In the context of the affected endocrine functions, we identified a group of genes related to *aldosterone metabolic process* and renin-angiotensin-mediated blood flow regulation (*regulation of systemic arterial blood pressure by mediated by a chemical signal*), including *Ace2*[115,116] that has negative effects on glucose tolerance and insulin sensitivity[117], or *Adrb2* which impairs endocrine development by generating hypervascularized islets[118]. Additionally, β-cell proliferation and insulin secretion regulator *Sgk3*[119] were upregulated in *Neurod1*ST.

Mammalian Phenotype Ontology enrichment analysis uncovered the enrichment of differentially expressed genes in *Neurod1*ST associated with phenotype-matching categories (Supplementary Data 1), such as *abnormal islet morphology* (downregulated genes) and *abnormal homeostasis* (upregulated genes). Interestingly, downregulated genes were enriched exclusively in pancreatic endocrine tissues, whereas upregulated genes were found in a wide span of multiple tissues, e.g., lung, renal, neural, dermal, and gastrointestinal, including pancreatic.

Next, using an open-access transcriptomic database of the mouse embryonic (E15.5) single-cell pancreas[29], we linked identified differentially expressed genes from our bulk-RNA-seq to their characteristic single-cell transcriptomic signature profiles in the cell subpopulations of the differentiating pancreas (Supplementary Data 1). We categorized differentially expressed genes based on their single-cell transcriptomic signature profiles as endocrine, non-endocrine, and combining both. Strikingly, we observed a clear pattern of non-endocrine genes upregulated in *Neurod1*ST (Fig. 4e), representing mesenchymal, ductal, trunk, and acinar cell subpopulations. Conversely, genes with an endocrine expression pattern were downregulated in *Neurod1*ST. Based on the functional characterization of differentially expressed genes using DAVID[120], non-endocrine genes were enriched in the cluster of *transcription regulators* (Fig. 4f), *cell differentiation* (Supplementary Fig. 4a), and *cell cycle* (Supplementary Fig. 4b) in *Neurod1*ST compared to the *Control*. Prominent endocrine differentiation-driving transcription factors *Insm1*[121,122], *Isl1*[123], *Mafa*[124], and *Tox3*[125], including *Neurod1*, were downregulated in *Neurod1*ST as well as chromatin remodelers *Kdm1b*[126], and α-cell enriched *Smarca1*[127]. Additionally, the β-cell apoptosis regulator *St18*[128], endocrine β-cell

survival regulator *Slit1*[129], and acinar-to-ductal inductor *Klf4*[130] were decreased. In contrast, various transcription factors were upregulated in *Neurod1*ST such as ductal *Sox9*[92] and its cooperator *Maf*[131,132], acinar-reprogramming *Mecom*[111], or transcriptional corepressor *Cbfa2t3*[133], and positive proliferation marker *Mki67*, cyclin *Ccnb2*, negative proliferation regulator *Zfp36l1*[134], β-cell protective regulator *Rgs2*[135], or ductal cell cycle-driver *Id3*[108]. Two chromatin remodelers were also upregulated; the ubiquitous *Smarca4* and EPs-distinct *Smarcd2*[125].

To better understand the ambiguous cell identities in *Neurod1*ST, we performed the deconvolution of our bulk-RNA-seq data to estimate cell type proportions[136]. Single-cell transcriptomic signatures of five mostly abundant (at E15.5) and closely cell-fate-related pancreatic cell types (three endocrine and two non-endocrine), namely α cells, β cells, endocrine progenitors, and trunk epithelium, and tip epithelium were used for the deconvolution analysis[137]. We identified a shift in the transcriptome signatures of β-cell populations between the *Neurod1*ST and *Control* (Fig. 4g).

Selected differentially expressed genes were validated by qRT-PCR with RNA isolated from whole E15.5 embryonic pancreas (Fig. 4h). However, validation of differentially expressed non-endocrine genes was not possible due to their abundance in non-endocrine tissue, which would greatly skew the expression differences. Apart from supporting the conclusions of disrupted endocrine functions (*Cpe, Resp18, Scg5, Slc2a5*) and EP-related transcription dysregulation (*Tox3, Smarcd2*), we also confirmed that *Neurod1* elimination affects its downstream target *Nnat*[138]. Furthermore, we used immunohistochemistry to assess the expression of the non-endocrine ductal SOX9 transcription factor (Fig. 4i, j). Consistent with our bulk-RNA-seq data, we observed frequent co-expression of SOX9 and PDX1 in tdTomato⁺ endocrine cells in *Neurod1ST*, whereas such cells were rarely detected in the *Control* pancreas at E15.5.

Next, we searched for transcription factor binding motifs enriched within the promoters of all differentially expressed genes found in the *Neurod1*ST mutant, using HOMER de novo motif analysis[139]. This analysis identified significant enrichments for binding sites of NEUROD1 and ISL1 within the extracted top ten hits for the promoters of down-regulated genes (Fig. 4k), confirming direct gene expression network regulation involving both downregulated transcription factors. Interestingly, the motifs of lineage-determining transcription factors, such as MEIS2, NFIA and RBPJ, were also significantly enriched in the promoters of down-regulated genes. RBPJ[140,141] is a crucial modifier of endocrine-versus-acinar transcriptional activity and both RBPJ and MEIS2[142] modulate PDX1 activity in a lineage-specific manner, while NFIA regulates Notch signaling and promotes pro-endocrine development[143]. Among motifs enriched in differentially expressed upregulated genes, we identified HNF4α, which is a master regulator of epithelial differentiation in multiple tissues, including the gastrointestinal tract and liver[144,145]. In the pancreas, HNF4α controls β-cell function and glucose metabolism and is associated with MODY1[146,147]

but HNF4α also regulates early pancreatic genes in the pancreatic progenitors[148]. Additionally, significant enrichments for binding sites of EGR1, HES1, ELF5, HIC1, SPDEF, and NR2E1 transcription factors were identified in upregulated gene promotors. EGR1 regulates PDX1 expression in β cells[149] and was also described as their stress marker[150]. HES1 controls cell fate determination between ductal-versus-endocrine fate choice in bipotent progenitors via the Notch signaling axis[12,151]. ELF5 is specifically expressed in trunk cells, while SPDEF and HIC1 in acinar cells at E15.5[29]. On the other hand, NR2E1 possess β-cell protective traits[152].

Taken together, these data identified significant downregulation of endocrine genes on the global transcriptomic level within the Neurod1-deficient endocrine cell population, while the expression of numerous genes characteristic for the non-endocrine tissues is substantially elevated. These deviations fully support the phenotypic

pathophysiology of Neurod1ST and emphasize the flawed developmental gene regulatory network derailing endocrine differentiation and cell cycling. Furthermore, these results provide further evidence of the essential role of NEUROD1 in defining basal characteristics of the endocrine lineage commitment.

## NEUROD1 shapes the endocrine cell epigenetic landscape

Following the findings of NEUROD1 pioneering potential[31,33,34], we decided to further explore its regulatory role in the endocrine identity determination of the developing pancreas using CUT&Tag sequencing (Fig. 5a). To assess NEUROD1-related changes in bivalency, we compared the H3K4me3 and H3K27me3 endocrine profiles of Neurod1ST-Ai14 with Control-Ai14. Both profiles were prepared from FACS-sorted tdTomato+ endocrine cells isolated from several E15.5 embryonic pancreases. To correlate H3K4me3 and H3K27me3 modifications with

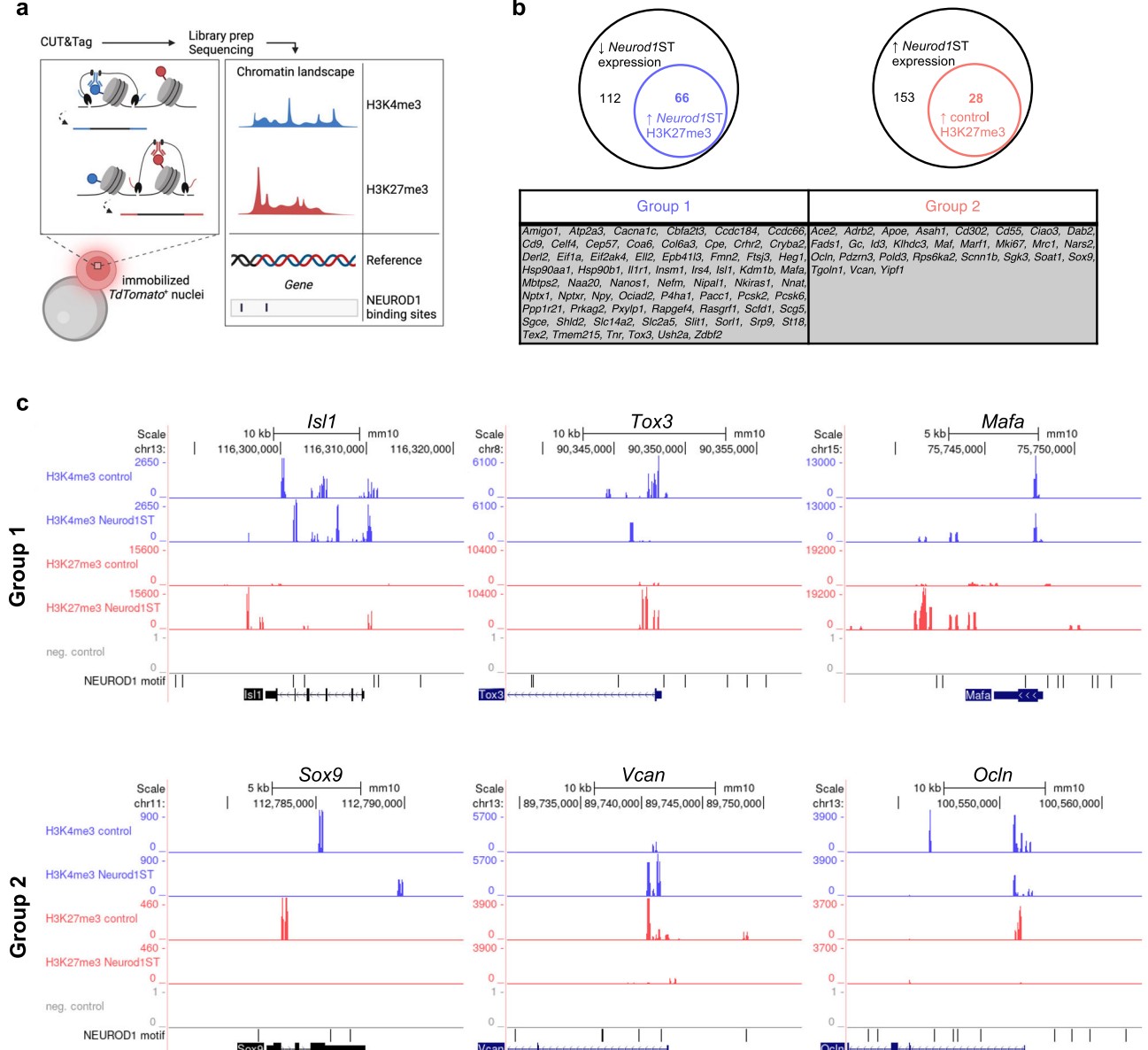

**Fig. 5 | NEUROD1 is essential for shaping the epigenetic landscape distinct to the endocrine cell lineage. a** CUT&Tag-seq experiment design: tdTomato+ nuclei were immobilized on magnetic beads and "Cleavage Under Targets and Tagmentation" was performed with antibodies against H3K4me3 or H3K27me3. Reads from amplified libraries were mapped on reference genome and putative NEUROD1 binding sites were identified in silico. Created with BioRender.com. **b** Among 112

downregulated genes in Neurod1ST, we found 66 genes containing elevated H3K27me3 peaks (Group 1). In cohort of 153 upregulated genes in Neurod1ST, we identified 28 genes (Group 2) with increased H3K27me3 peaks in Control compared to Neurod1ST. **c** Representative genes for both Group1 and 2 are presented as an UCSC genome browser output. NEUROD1 binding motifs identified in silico are often adjacent to the detected peak loci of gene promoters.

gene expression changes, we focused only on histone modifications of differentially expressed genes identified by our bulk-RNA-seq. We also included NEUROD1 binding sites identified in silico by HOMER[139]. As either the presence of the repressive H3K27me3 mark[153] or the combination of H3K27me3 with H3K4me3 histone modifications[154] result in downregulation of gene expression or silencing, we centered our scope on H3K27me3 peaks that might possibly cause gene downregulation.

Based on the H3 methylation pattern, two main gene groups were identified (Fig. 5b). Group 1 contained genes downregulated in *Neurod1*ST-*Ai14* with increased H3K27me3 peaks within the gene promoter, gene-adjacent loci, or both. Many identified genes were directly linked to endocrine functions by our analysis of bulk-RNA-seq data, indicating *Neurod1*-dependent modulation of the gene regulatory network driving endocrine differentiation, e.g., *Isl1*, *Tox3*, *Mafa* (Fig. 5c). Group 2 consisted of genes containing upregulated H3K27me3 peaks in the *Control* samples, such as *Sox9*, *Vcan*, *Ocln*, correlating with downregulated expression in endocrine cells of the *Control-Ai14* pancreas based on our bulk-RNA-seq data (Fig. 5b, c). In contrast, H3K27me3 modification was undetected in Group 2 genes in *Neurod1*ST endocrine cells, supporting a putative cause of their upregulated expression in *Neurod1*ST.

These findings indicate that NEUROD1 is a key player in the gene regulatory networks underlying cell fate commitment, defining endocrine progenitor identity by the remodeling of the chromatin landscape and thus, driving the terminal differentiation of endocrine cells in the pancreas. Furthermore, our results uncover a new point of view on the global scope of yet unclear epigenetic regulations behind such phenomena.

## Molecular reprogramming in *Neurod1*-deficient α and β cells

Although a bulk-cell RNA-seq provides extensive sequencing depth, it represents an average of gene-expression of multiple cells. Therefore, to examine the effects of *Neurod1* deletion in a heterogeneous endocrine population, we utilized single-cell RNA-seq (scRNA-seq) analysis. We specifically focused on adult islets of Langerhans to minimize a possible asynchrony of generation of endocrine cells between the control and mutant pancreases. For the scRNA-seq study, we used a conditional *Neurod1* deletion model (*Neurod1CKO*; *Neurod1*^loxP/loxP^;*Isl1*^Cre/+^ genotype)[45]. Unlike our *Neurod1*ST mice that die shortly after birth, 40% of *Neurod1CKO* mice survive into adulthood. Isolated pancreatic islets were dissociated into single cells, and an estimated 12,000 cells per each genotype were used for scRNA-seq. Unsupervised clustering of cell endocrine profiles revealed six transcriptionally distinct subgroups in the *Control* (Fig. 6a). Four principal clusters were identified based on endocrine markers such as *Ins2* and *Ins1* for the β cell population, *Gcg* for α cells, pancreatic polypeptide-expressing cells (*Ppy*) for PP/γ-cell population cluster, and *Sst*-expressing cells (δ cells). Some cells within the δ cell cluster co-expressed *Neurog3* at a low level[155]. Notably, many β cells co-expressed *Sst* and *Ins2* and/or *Ppy* in *Neurod1CKO*. In addition, we found a higher proportion of polyhormonal cells in *Neurod1*CKO compared to the *Control* clusters (Fig. 6b).

Our scRNA-seq of islets of Langerhans of *Neurod1CKO* revealed a remarkable difference in the β-cell population, which segregated into two molecularly distinct subpopulations. One β-cell subpopulation of *Neurod1CKO* overlapped with the β-cell population in *Control* animals in the UMAP space, suggesting that this β-cell subpopulation, which shares molecularly similarities with the *Control* β-cell subpopulation, is responsible for the survival of *Neurod1CKO* mutants. The second β-cell subpopulation exhibited a loss of *Ins1* and *Neurod1* and therefore, was further investigated to gain insights into the molecular characteristics of β cells lacking *Neurod1*. The relative proportions of endocrine subtypes were altered in *Neurod1CKO* with a reduced proportion of α and PPY cells, and increased proportions of β-cell subpopulations

compared to endocrine subpopulations in the *Control* (Fig. 6b). A noticeable shift in the distribution of α-cell clusters indicates the molecular identities of α cells were also different between *Neurod1CKO* and *Control* (Fig. 6c). Other mutant endocrine cells were clustered into the similar subgroups as those observed in the *Control* in the UMAP space (Fig. 6c).

We next compared the gene profiles of *Neurod1*-negative cell subtypes in *Neurod1CKO* with *Control* endocrine cells. To reveal changes in expression patterns underlying the distinct molecular shifts of α- and β-cell subpopulations in *Neurod1CKO*, we performed a differential gene expression analysis. By comparing the single-cell transcriptomes of *Neurod1CKO Neurod1*^negative^ α and β cells with *Neurod1*^positive^ cells of the *Control*, we observed clear differences in gene expression (Fig. 6d, Supplementary Data 2), indicating that *Neurod1* deletion had a significant impact on the global expression profiles of both endocrine cell subtypes.

The most significant transcriptional differences were observed between β cells of the *Control* and *Neurod1CKO*. In the *Neurod1CKO* β-cell subpopulations, we observed a downregulation of genes associated with the insulin secretion machinery and hormone production, such as *Ins1*, *Ins2*, *G6pc2*, and *Ffar1*. These findings are consistent with the gene expression changes we detected in our bulk-RNA-seq analysis of E15.5 endocrine cells. Importantly, our scRNA-seq analysis focused on adult β cells, which allowed us to better discern the molecular differences between the altered matured, immature, and healthy β-cell states. Surprisingly, the expression of key maturation markers of β cells such as *Slc2a2* and *Ucn3*[156,157] in *Neurod1CKO* was comparable to *Control* β-cell subpopulations, except for the absence of *Ucn3* in a smaller subcluster of adult β cells in *Neurod1CKO* (indicated by the arrowhead in Fig. 7). However, there was a notable increase in the expression of an immature β-cell marker, *Mafb* (Fig. 7b). *Mafb* is a transcription factor that serves as an immature β-cell marker highly expressed during the development of pancreatic α and β cells, but its expression becomes restricted to α cells in adult mice[158,159]. Other upregulated markers of immature β cells included retinol binding protein 4 (*Rbp4*)[62] and a pan-endocrine lineage marker gene, *Chga*[125]. *Rbp4*-rich cells in the adult pancreas represent dedifferentiated β cells with reduced functionality[157,160]. Our analysis revealed an enrichment of *Wnt4* in *Neurod1CKO*, which may indicate an adaptive response by the β cells to enhance their metabolic maturation[161] (Fig. 7). Additionally, we observed the upregulation of key transcription factors for insulin gene expression and β-cell survival, *Isl1*, *Nkx6.1*, and *Pdx1*[61,162–164]. These finding suggest a compensatory response by the dysfunctional β cells in *Neurod1CKO* to overcome the absence of *Neurod1* and maintain their survival and functional integrity (Fig. 7 and Supplementary Fig. 5b). It is interesting to note that this contrasts with previous reports where these transcription factors were significantly reduced in diabetic islets, indicating a loss of β-cell identity under glucotoxic conditions[157,165,166]. Nevertheless, metabolic stress in failing pancreatic β cells in *Neurod1CKO* was evident by the upregulation of metabolic genes associated with glutathione and lipid metabolism, glycolysis, and protein processing in the ER. These included lactate dehydrogenase A (*Ldha*), free fatty acid receptor (*Ffar4*), mitochondrial fatty acid beta-oxidation enzyme (*Hadh*), the ER-resident sterol O-acyltransferase (*Soat1*), and glutathione peroxidase 1 and 3 (*Gpx1*, *Gpx3*) (Fig. 7, Supplementary Fig. 5b, and Supplementary Data 2).

Consistent with the findings from our bulk-RNAseq analysis of E15.5 endocrine cells, our single-cell transcriptomic analysis confirmed the upregulation of non-endocrine genes in *Neurod1CKO* adult endocrine cells. These included the ductal marker *Sox9*[92], the WNT secretion mediator *Wls*[93,94], *Ocln*[96], *Cdh4*[103], and the acinar-reprogramming factor *Mecom*[111] (Fig. 7, Supplementary Fig. 5b). These expression changes suggest a shift in the cellular identity and a compromised commitment to the endocrine lineage in the *Neurod1CKO* adult endocrine cells.

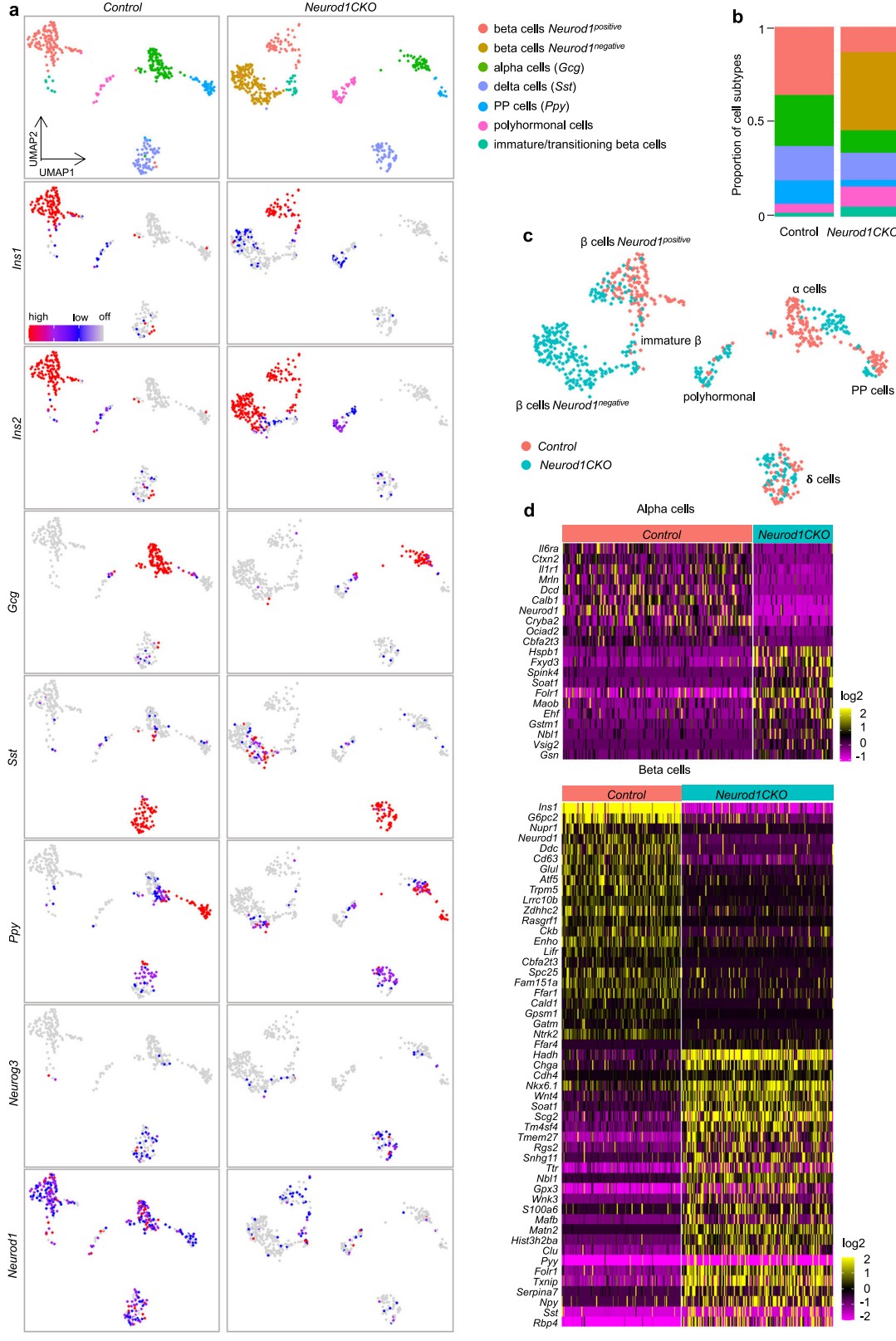

**Fig. 6 | Molecular identity of the β-cell subpopulation is altered by *Neurod1* deletion. a** Two-dimensional embedding and unsupervised clustering of endocrine profiles from *Control* pancreatic islets revealed six cell clusters, identified by markers shown in a cropped and transformed UMAP plots: β cells (*Ins1, Ins2*), α cells (glucagon, *Gcg*), PP cells (pancreatic polypeptide, *Ppy*), polyhormonal cells (expressing *Ins1, Ins2, Gcg, Sst*, or/and *Ppy*), immature/transitioning β cell cluster (*Ins1, Ins2*), and δ cells (*Sst*). The detected clusters are indicated by different colors.

In *Neurod1CKO*, endocrine profiles revealed the same six cell clusters with one additional cluster representing a molecularly different β-cell subpopulation. **b** Cellular composition per sample. **c** UMAP plot showing the distribution of endocrine cells from the *Control* and *Neurod1CKO*. **d** Heatmaps of differentially expressed genes between *Control* and *Neurod1CKO* in α and β cells (cut off: $P_{adj} < 0.05$, log2FC = 0.6).

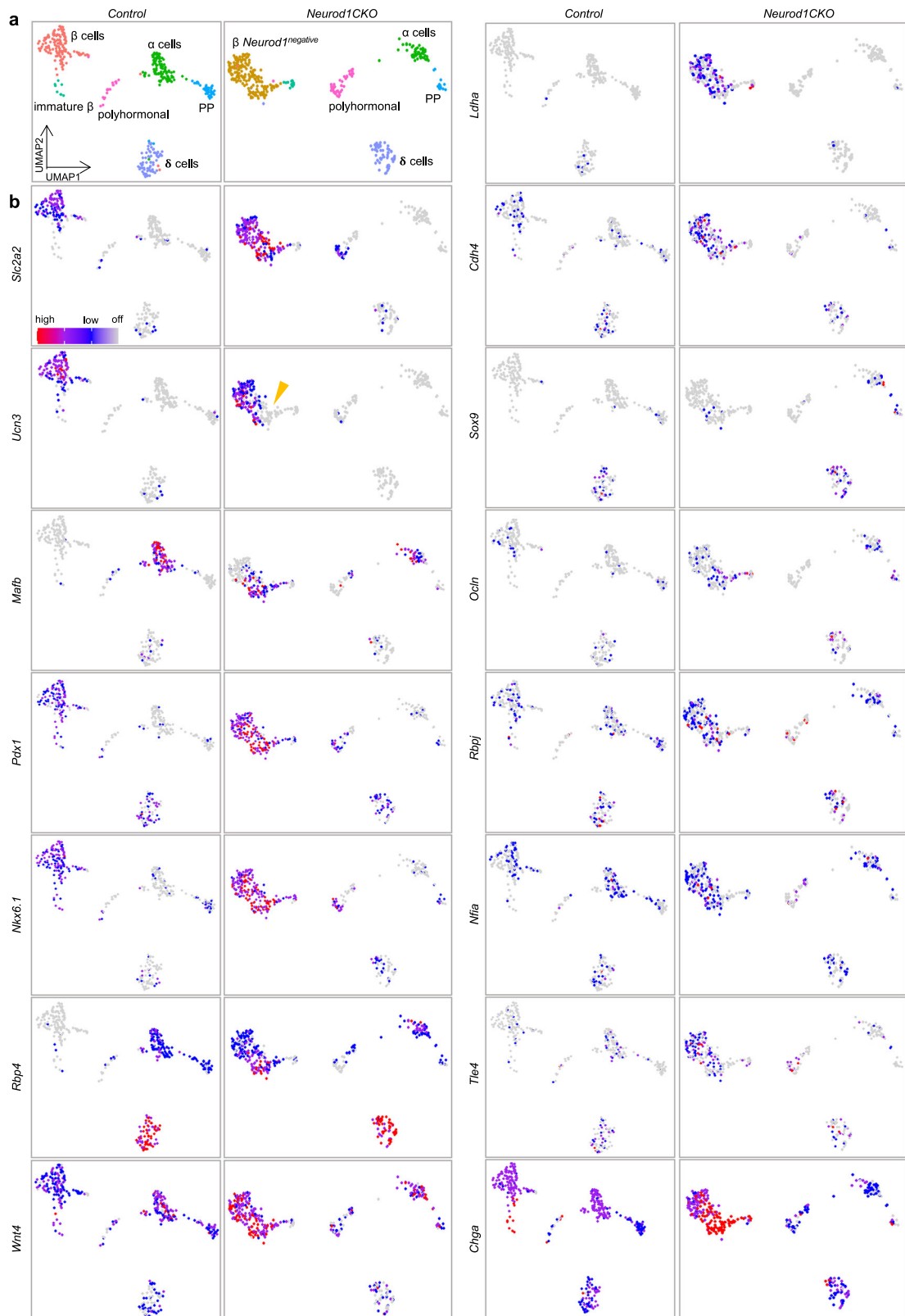

**Fig. 7 | Molecular reprograming of *Neurod1CKO* endocrine cells. a** UMAP visualization (in two dimensions [UMAP_1 and UMAP_2]) of endocrine cells from the adult *Control* and *Neurod1CKO* pancreas (different colors mark distinct cell types annotated based on the expression of signature genes; each dot represents one cell). Clusters corresponding to α, β, PP (γ), δ, polyhormonal cells, immature β, and β cells not expressing *Neurod1* (in *Neurod1CKO*) are shown in cropped and transformed UMAP plots that excludes a cluster of β cells expressing *Neurod1* in *Neurod1CKO* plots for easier visualization. **b** Feature plots showing differentially expressed genes between *Control* and *Neurod1CKO* endocrine cells.

Remarkably, we identified an upregulation of lineage-determining factors and regulators of Notch-signaling regulators, *Nfia*[143] and *Rbpj*[140,141], in *Neurod1CKO* β cells (Fig. 7). A significant enrichment of NFIA and RBPJ binding sites within the promoters of downregulated genes from the bulk-RNAseq data was revealed in our HOMER analysis (Fig. 4k). These findings further support the involvement of these factors in the transcriptional dysregulation observed in the *Neurod1CKO* β cells and suggests their potential role in mediating the downstream effects of *Neurod1* deficiency.

Furthermore, our scRNA-seq analysis uncovered changes in gene expression patterns in *Neurod1CKO* that have not yet been investigated in the context of developing endocrine cells, dysfunctional β cells, or diabetic islets. For example, we found the enrichment of the transcriptional corepressor *Tle4*, which has been implicated in various pathways and transcription factor interactions, including Wnt and Notch signaling[167]; upregulation of citron rho-interacting kinase (*Cit*), which is known to play roles in cell division, proliferation, and neuronal development[168]; and increased levels of the transcription factor *Nfix* that plays multiple roles in cell fate determination, differentiation, cell migration, and oxidative stress[169–171]. Conversely, we observed a marked reduction in the expression of the cadherin gene *Cadh8* in *Neurod1CKO*.

Altogether these findings provide additional evidence of cellular reprogramming and dysregulation occurring at the single-cell level within both α- and β-cell populations and offer valuable insights into the molecular mechanisms underlying the deficiency of *Neurod1*.

## Discussion

In this study, we identified a previously unrecognized function of NEUROD1 during the differentiation of pancreatic endocrine cells. Our results suggest that NEUROD1-induced changes in the transcriptome and epigenome reinforce endocrine cell fate commitment and promote endocrine differentiation (Summary in Fig. 8). We demonstrated that NEUROD1 is not required for α-cell generation during the primary transition of pancreas development, as the percentage of generated GCG-producing α cells during the primary transition are comparable between *Neurod1*ST and *Control*. Although the "first wave" of α-cell generation seems to be unaffected in *Neurod1*ST, many of these cells co-expressed PDX1, indicating abnormalities in the pancreatic lineage differentiation pathway. Consistent with previous studies[44,45], we confirmed an impaired proliferative expansion of α and β cells lacking NEUROD1 in the *Neurod1*ST perinatal pancreas. Although previous studies suggested that NEUROD1 is a dominant factor for the maturation of β cells and *Ins1* expression[44,45,48], there are no mechanistical investigations to date

on NEUROD1 requirements for differentiation programs of endocrine cells.

Our transcriptomic analyses of endocrine cells investigated the secondary transition of endocrine differentiation at E15.5, when endocrine progenitors have a higher potential to form β cells[125]. At E15.5, most endocrine progenitors come from the bipotent trunk epithelium, and it is plausible that the differentiation of endocrine progenitors is reinforced by changes in the transcriptome and epigenome. Accordingly, *Neurod1* elimination in *Neurod1*ST resulted in downregulation of transcription factor networks driving endocrine differentiation, maintaining processes associated with endocrine functions, and insulin expression, including *Insm1*, *Isl1*, and *Mafa*[49,122,124,162]. In addition, the expression of chromatin remodelers was changed in *Neurod1*ST endocrine cells, the ubiquitous *Smarca4* and EPs-distinct *Smarcd2* were upregulated, while *Kdm1b*[126], and α-cell enriched *Smarca1*[127] were downregulated. Strikingly, we identified the upregulation of non-endocrine genes in *Neurod1*ST, including ductal *Sox9*[92] and its cooperator *Maf*[131,132], or acinar-reprogramming *Mecom*[111]. The characterization of enriched transcription factor binding motifs within promoters of downregulated genes showed a distinct enrichment for endocrine lineage-determining transcription factors, NFIA and RBPJ[140,141,143], and endocrine lineage specific transcription factors, ISL1 and NEUROD1, correlating with *Neurod1* deficiency and changes in transcription signatures of *Neurod1*ST endocrine cells. Among enriched transcription factor binding motifs in the upregulated gene promoters were the Notch effector HES1 that controls cell fate bipotent progenitors[151], ELF5 associated with the trunk epithelium of the pancreas[29], and HNF4a important for the expression of early pancreatic progenitor genes[148]. These results suggest that NEUROD1 reinforces the downstream transcriptional network of bipotent progenitor cells toward endocrine lineage differentiation preferentially towards β cells in the secondary transition of the E15.5 pancreas.

Like the transcriptome changes, we found a corresponding shift in the chromatin landscape of H3K4me3 and H3K27me3 modifications in *Neurod1*ST pancreatic endocrine cells. The silencing H3K27me3 modification was detected in the gene promoter regions of *Isl1*, *Insm1*, and *Mafa* genes corresponding to the downregulation of mRNA of these genes in *Neurod1*ST. In contrast, the upregulation of non-endocrine mRNA, *Sox9*, *Vcan*, and *Ocln*, in *Neurod1*ST was associated with a missing H3K27me3 silencing modification in *Neurod1*ST compared to *Control* endocrine cells. These data support an epigenetic regulatory role of NEUROD1 in the differentiating endocrine cells in the pancreas and highlight more robust requirements of NEUROD1 for the formation of fully functional endocrine cells.

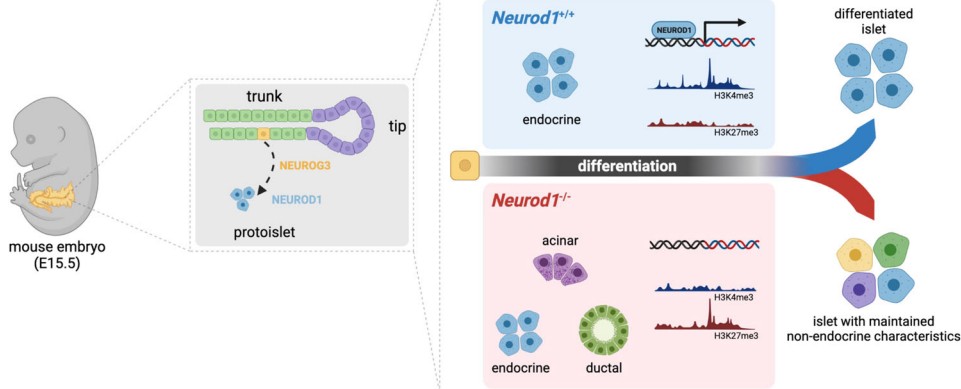

**Fig. 8 | NEUROD1 function during endocrine cell differentiation.** Our study reveals dysregulated transcriptome and altered histone methylation patterns in *Neurod1*ST endocrine cells. Compared to the *Control* islets of Langerhans, the *Neurod1*ST endocrine cells exhibit reduced expression of endocrine-related genes and enrichment of non-endocrine genes associated with the exocrine system and pancreatic ductal cells. These findings emphasize the critical role of NEUROD1 in governing gene networks that determine the commitment and terminal differentiation of endocrine cells in the pancreas. Created with BioRender.com.

In line with our bulk RNA-seq and epigenetic analyses, the single-cell profiling of endocrine cells from the adult pancreatic islets demonstrated that β-cell identity and function were dramatically altered by the embryonic deletion of *Neurod1*. The transcriptional β-cell states of *Neurod1CKO* represented a mixture of upregulated immature (*Mafb, Chga, Wls, Nfia, Rbpj, Ldha*) and mature β-cell marker genes (*Slc2a2, Ucn3*) together with upregulated genes representing non-endocrine fate choices (*Sox9, Cdh4, Ocln, Mecom*). Interestingly, we found a higher proportion of β cells co-expressing *Sst* and *Ins2* and/or *Ppy*, as well as a higher proportion of polyhormonal cells within the *Neurod1CKO* endocrine subpopulations. The ability of NEUROD1 to govern cellular and fate choice identities is not without precedent. Several studies highlighted the significance of NEUROD1 in promoting neuronal fate and preserving neuronal identity by suppressing non-neuronal fates[33–35].

In the context of β-cell identities, *Neurod1* deficiency affected numerous genes that serve as markers for β-cell dysfunction and the diabetes-induced dedifferentiation of β-cells[157,160]. These genes play crucial roles in insulin secretion, metabolism, and maturity. For instance, RBP4 is known as a diabetogenic factor[157,160] but is also expressed in immature β-cells during embryonic development[62]. Likewise, LDHA, a glycolytic enzyme, is normally suppressed in mature β-cells since it is critical for coupling glucose metabolism with insulin secretion. In contrast to reports of diabetes-induced dedifferentiated β cells[157,166], we found an increased expression of transcription factors associated with β-cell identity and the regulation of insulin genes, such as *Pdx1, Isl1*, and *Nkx6.1*, indicating a compensatory regulatory response aimed at overcoming the absence of *Neurod1*. Similarly, an enrichment of *Wnt4* and *Ffar4* in *Neurod1CKO* may indicate an adaptive response by the dysfunctional β cells to enhance their metabolic maturation and stimulate failing insulin secretion[161,172].

Recent findings have provided compelling evidence supporting the existence of transcriptional and functional heterogeneity among insulin-secreting β cells within the context of normal islet homeostasis. This heterogeneity encompasses a range of cellular states and sub-populations, highlighting the diverse characteristics of these cells[62,136,160]. Epigenetic silencing represents an important axis of β-cell heterogeneity[173]. Importantly, under conditions of metabolic stress, these subpopulations undergo dynamic changes that lead to functional impairments. An intriguing area for future investigation is whether *Neurod1*-deficient β cells, which exhibit a mixture of mature, immature, and non-endocrine identities, represent a subpopulation that possesses inherent plasticity and adaptive capacity to sustain chronic metabolic stress. Understanding such a heterogeneous transcriptional regulatory landscape, featuring rigidity and flexibility in expression across different gene families, would provide valuable insights into the functional diversity and adaptability of β cells.

Based on our transcriptomic and epigenetic analyses, we propose two regulatory roles of NEUROD1. Firstly, NEUROD1 controls the functional differentiation of insulin-producing β cells. Despite the perturbation of the β-cell differentiation program at the endocrine progenitor stage due to *Neurod1* loss, adult *Neurod1*-deficient β cells of *Neurod1CKO* still express key transcription factors associated with β-cell identity and insulin gene transcription, such as *Pdx1, Isl1*, and *Nkx6.1*. Our investigation of the maturation characteristics of adult β cells in *Neurod1CKO* revealed a combination of immature and mature states. This is evident from the enrichment of glycolytic genes (*Ldha* and *Eno1*), the pan-endocrine gene (*Chga*), and immature β-cell markers (*Mafb* and *Rbp4*), which resemble immature fetal or neonatal β cells. Interestingly, the expression of mature β-cell marker genes (*Slc2a2* and *Ucn3*) in *Neurod1CKO* was similar to the *Control*, while the upregulation of *Wnt4* suggests an attempt to enhance β-cell functional maturation.

Secondly, NEUROD1 plays a role in reinforcing endocrine commitment during the differentiation of endocrine progenitors. This is supported by the upregulation of non-endocrine genes, including *Sox9, Cdh4, Ocln*, and *Mecom*, observed in *Neurod1*ST. Endocrine cells primarily originate from Sox9⁺ bipotent progenitors in the trunk epithelium[174]. These progenitors either differentiate into endocrine cells or ducts[12]. SOX9 triggers endocrine differentiation by inducing *Neurog3* expression[92,174], and *Neurod1* expression follows *Neurog3*[25–28]. NEUROD1 expression was also found in transitioning Sox9low cells[43]. Furthermore, ectopic expression of NEUROD1 induces endocrine differentiation in the pancreas, demonstrating its pro-endocrine role[25]. Similarly, in *Neurod1CKO* adult β cells, there is an upregulation of non-endocrine genes, indicating permanent changes in β-cell identities. Moreover, the increased proportions of bihormonal and poly-hormonal β cells in *Neurod1CKO* suggest a potential involvement of NEUROD1 in controlling monohormonal β-cell state.

Taken together, our findings provide insights into the regulatory roles of NEUROD1 in promoting the functional differentiation of β cells. Our data suggest that NEUROD1 reprograms the transcription factor and epigenetic landscapes, favoring the differentiation of endocrine cells and facilitating the generation of functional insulin-producing β cells. Our study, combined with the well-established role of NEUROD1 in neuronal differentiation, emphasizes the broader significance of NEUROD1 as an epigenetic and transcriptional regulator in embryonic development.

### Limitations of the study
One limitation of our analysis is the lack of spatial resolution. By single-cell dissociation, information regarding intra-islet connectivity, the distribution of β-cell subtypes, and regional functional differences within the pancreas or islets is lost. Additionally, we did not explore the temporal gene expression kinetics, which represents another important aspect that could enhance our understanding of pancreatic development. Investigating the dynamic changes in gene expression patterns over time would provide insights into the molecular events and regulatory mechanisms involved in the formation of the *Neurod1*-deficient endocrine pancreas. The study also does not identify direct NEUROD1 target genes during the formation of endocrine cells in the pancreas. Finally, the main conclusions of the current study are based on a mouse in vivo model, although the findings should translate to human pancreas development, this has not been confirmed.

## Methods
### Ethics approval and consent to participate
Experiments were carried out following the animal welfare guidelines 2010/63/EC of the European Communities Council Directive, agreeing with the Guide for the Care and Use of Laboratory Animals (National Research Council. Washington, DC. The National Academies Press, 1996). The design of experiments was approved by the Animal Care and Use Committee of the Institute of Molecular Genetics, Czech Academy of Sciences (protocol # 878/2022).

### Experimental model
Animal experiments were conducted according to protocols approved by the Animal Care and Use Committee of the Institute of Molecular Genetics, Czech Academy of Sciences. All experiments were performed with littermates (males and females) cross-bred from two transgenic mouse lines: floxed *Neurod1* [*Neurod1*loxP/loxP, 58], and *Neurod1*Cre/+ [Tg(Neurod1-Cre)1Able, Stock No: 028364 Jackson Laboratory[59]]. The lines were maintained on the C57BL/6 background. *Neurod1*Cre/+ mice do not have any detectable phenotype. Breeding scheme: Female mice *Neurod1*loxP/loxP were crossed with *Neurod1*loxP/+; *Neurod1*Cre/+ males, in which the *Neurod1-Cre* knock-in allele was inherited paternally to minimize the potential influence of maternal genotype on the developing embryos. *Neurod1*loxP/+ or *Neurod1*loxP/loxP without the *Neurod1-Cre* allele individuals were used as the *Control*. The *Het* mice of the *Neurod1*loxP/+; *Neurod1*Cre/+ genotype, containing

one functional *Neurod1* allele, were functionally and phenotypically similar to the *Control*. The reporter line (Ai14, B6.Cg-*Gt(ROSA) 26Sor^tm14(CAG-tdTomato)Hze*, Stock No: 7914 Jackson Laboratory) was used, generating *Control-Ai14* (*Neurod1^loxP/+*; *Neurod1^Cre/+*; *tdTomato^loxP/+*) and *Neurod1ST-Ai14* (*Neurod1^loxP/loxP*; *Neurod1^Cre/+*; *tdTomato^loxP/+*). For scRNA-seq, we used mice with a conditional deletion of *Neurod1*, *Neurod1CKO*, with *Neurod1^loxP/loxP;Isl1^Cre/+* genotype[45]. The mice with the genotype *Neurod1^loxP/+* or *Neurod1^loxP/loxP* without the *Isl1-Cre* allele were used for the preparation of the *Control* pancreatic cells. Genotyping was performed by PCR on tail DNA (Supplementary Table 1). Mice were kept under standard experimental conditions with a constant temperature (23–24 °C) and fed the standard diet (#1324, Altromin International, Germany). The females were housed individually during the gestation period, and the litter size was recorded. Blood glucose levels were measured in animals by glucometer (COUNTOUR TS, Bayer); blood glucose levels maintained above 13.9 mmol/L are classified as diabetic. For total pancreatic insulin content, pancreases were excised, weighed, minced, and homogenized in acid-ethanol. Hormone concentration in extracts was measured by ELISA using Mouse Insulin ELISA kit (Mercodia, Sweden).

## Reverse transcription-quantitative real-time polymerase chain reaction

RT-qPCR was performed as described previously[175]. Briefly, total RNA was isolated from the whole pancreas at E15.5 by Trizol RNA extraction. The *Hprt1* gene was selected as the best reference gene for our analyses from a panel of 12 control genes (TATAA Biocenter AB, Sweden). Primers were designed using Primer-BLAST[176]. Primer sequences are presented in Supplementary Table 2.

## Immunohistochemical staining and morphological evaluations

For vibratome sections, dissected tissues were fixed in 4% PFA, embedded in 4% agarose gel and sectioned at 80 μm on a Leica VT1000S vibratome. Used primary and secondary antibodies are in Supplementary Table 3. The nuclei were in specific cases counterstained with Hoechst 33342. Apoptotic cells were labeled by TUNEL (#11684795910, Roche). Image acquisition was completed using the Zeiss LSM 880 NLO scanning confocal microscope with ZEN lite program. The expression of NEUROD1 was quantified in the whole pancreases of *Control* and *Neurod1ST* at E10.5 using the thresholding tool of the ImageJ, and the results were expressed as a percentage of the NEUROD1+ area per tdTomato+ area. The numbers of GCG+, INS+, and Ki67+ cells were counted in all viewing areas of a vibratome section from the central part of the pancreas with the largest pancreatic footprint from *Neurod1ST* and *Control* embryos or pups. The results for GCG proliferating and INS proliferating cells were expressed as a percentage of the total number of GCG+ and INS+ cells per evaluated area of the pancreas. The Cell Counter plugin of ImageJ[177] was used for quantification. GCG+ and INS+ cells were counted in a vibratome section from the central part of the pancreas with the largest pancreatic footprint (*n* = 5 pancreases/genotype in total 24 areas of view/*Control* and 22/areas of view/*Neurod1ST*). The expression of SST was quantified in all viewing areas of a vibratome section from the central part of the pancreas with the largest pancreatic footprint in *Neurod1ST-Ai14* and *Control-Ai14* pups. The thresholding tool in ImageJ was utilized for quantification, and the results were expressed as a percentage of SST+ areas per the total area of tdTomato+ per evaluated area of the pancreas. The evaluation of GCG+ delaminating cells at E10.5 was quantified using the thresholding tool in ImageJ and the results were expressed as a percentage of the total GCG+ area to PDX1+ area. GCG+ cells co-expressing PDX1 were quantified in the entire pancreas at E10.5. Quantification was performed using the Cell Counter plugin of ImageJ, the results were expressed as a percentage of GCG+ PDX1+ cells per total GCG+ cells. The thresholding tool in ImageJ was utilized for the evaluation of tdTomato+ cells, which did not express GCG or INS in the *Neurod1ST-Ai14* and *Control-Ai14* pancreas at E15.5. The expression of tdTomato was quantified in all viewing areas of a vibratome section from the central part of the pancreas with the largest pancreatic footprint. The results were expressed as a percentage of GCG^negative and INS^negative areas per the tdTomato+ areas.

## Light-sheet fluorescent microscopy (LFSM) and analysis of images

The pancreas was microdissected from *Control-Ai14* and *Neurod1ST-Ai14* embryos (E18.5) and tissue was processed as described previously[178]. In addition to tdTomato expression, cleared samples were immunolabeled using anti-insulin and anti-GLP1 antibodies (Supplementary Table 4). Zeiss Lightsheet Z.1 microscope with illumination objective Lightsheet Z.1 5x/0.1 and detection objective Dry objective Lightsheet Z.1 5x/0.16 was used for imaging at the Light Microscopy Core Facility of the Institute of Molecular Genetics of the Czech Academy of Sciences. IMARIS software v8.1.1 (Bitplane AG, CA, USA) was used for image processing.

## Isolation of pancreatic endocrine cells

The pancreases microdissected from E15.5 embryos were directly trypsinized to generate single cells and prepared for FACS as described[61,179]. Cell suspensions were filtered through 40-μm nylon mesh and tdTomato+ cells were immediately sorted using a flow cytometer (BD FACSAria™ Fusion), through a 100 μm nozzle in 20 psi, operated with BD FACSDiva™ Software. For RNA sequencing, 100 cells per sample were collected into the individual wells of 96-well plate containing 5 μl of lysis buffer of NEB Next single-cell low input RNA library prep kit (New England Biolabs #E6420). For CUT&Tag profiling study, ~5000-20000 tdTomato+ cells per sample were sorted. The total time from euthanasia to cell collection was ~3 hrs. Cell sorting was performed in the Imaging Methods Core Facility at BIOCEV.

## Bulk-cell RNA sequencing and analyses

Using NEB Next single-cell low input RNA library prep kit for Illumina (New England Biolabs #E6420), RNA-seq libraries were prepared from the pancreases of *Neurod1ST-Ai14* (*n* = 4) and *Control-Ai14* (*n* = 4) by following the manufacturer's instructions. The libraries were sequenced on an Illumina NextSeq 500 next-generation sequencer. NextSeq 500/550 High Output kit 75 cycles (Illumina #200024906) were processed at the Genomics and Bioinformatics Core Facility (Institute of Molecular Genetics CAS, Czechia). RNA-Seq reads in FASTQ files were mapped to the mouse genome using STAR [version 2.7.0c[180]] GRCm38 primary assembly and annotation version M8. The raw data of RNA sequencing were processed with a standard pipeline. Using cutadapt v1.18[181], the number of reads (minimum, 23.4 million; maximum, 27.5 million) was trimmed by Illumina sequencing adaptor and of bases with reading quality lower than 20, subsequently reads shorter than 20 bp were filtered out. Ribosomal RNA and reads mapping to UniVec database were filtered out using bowtie v1.2.2. with parameters -S -n 1 and SortMeRNA database[182]. A count table was generated by Rsubread v2.0.1 package using default parameters without counting multi-mapping reads.

DESeq2 [v1.26.0[183]] default parameters were used to normalize data and compare the different groups. Genes were then filtered using the criteria: adjusted *P*-value $P_{adj} < 0.05$, and a base mean ≥ 50, and Fold change > 1.3 and <0.7. Only protein coding genes were kept in the analysis, following Mouse Genome Database[184]. The functional enrichment analysis was performed using: g:Profiler (ver. e106_eg53_p16_65fcd97) with g:SCS multiple testing correction method applying significance threshold of 0.05[71]; and MouseMine (http://www.mousemine.org/), using Mammalian Phenotype Ontology[185] and Mouse Anatomy and Development Ontology[186]. De novo motif analysis was performed by HOMER [ver. 4.10[139]] using default settings; up to top 5 scoring targets were considered per a detected motif.

## Deconvolution of endocrine cell subtypes for bulk-cell RNA sequencing data

Deconvolution was performed using the CibersortX algorithm at cibersortx.stanford.edu[187]. Single-cell transcriptomic profiling dataset of cells in the embryonic pancreas[137] was used as reference, including count matrix and metadata labels. The reference matrix was built out of the 2589 cells and gene list of 18565 gene features, as deposited by[137]. Every cell population counted over 250 cells. The units of reference matrix were UMI counts. Calculation of scRNA-seq signature matrix was done in default mode (quantile normalization disabled, minimal expression of 0.75, replicates of 5, sampling of 0.5). Imputation of cell fractions and high-resolution gene expression imputation were calculated in the default settings, with S-mode batch correction enabled, quantile normalization disabled and n = 100 permutations for significance analysis. Sample mixture file was submitted with unfiltered gene list of 26447 features for *Neurod1*ST and in UMI counts.

## CUT&Tag profiling

Bench top CUT&Tag version 3 was performed as previously described[57,188] with minor modifications[61]. Briefly, nuclei from freshly FACS-sorted tdTomato+ pancreatic endocrine cells were captured by Concanavalin A-coated magnetic beads to facilitate subsequent washing steps. CUT&Tag validated primary antibodies anti-H3K4me3, anti-H3K27me3, normal rabbit IgG negative control, and anti-rabbit secondary antibody were used (Supplementary Table 5). Each purified DNA library for Illumina sequencing was tested on Agilent High Sensitivity DNA Chip. DNA libraries were sequenced at the OMICS Genomics facility (BIOCEV) on the MiSeq (Illumina) using MiSeq Reagent Kit v3 (Illumina), which allows extended read lengths up to 2x 75 bp. Both CUT&Tag-seq. H3K27me3 and H3K4me3 data are from one independent biological replicate, each sample was pooled together from 3 to 7 pancreases.

Data analysis was performed following CUT&Tag Data processing tutorial[189]. Paired-end sequencing data were mapped using bowtie2 (ver. 2.2.5)[190] to mouse genome GRCm38 primary assembly. PCR duplicates were not removed. Reads mapping to SNAP-CUTANA barcodes were counted using bash shell tool grep. For visualization bam files were normalized using scale factor = "% genome reads" / "% spike-in reads" and converted to bigwig format using bedtools (ver. 2.30.0). Factors are available in Supplementary Table 6. After filtering and conversion to bedgraph format, peak calling was performed with usage of relevant IgG controls and stringent mode with tool SEACR (ver. 1.3)[191]. Peaks were annotated using CHIPseeker (ver. 1.30.3)[192] and annotation version M8 of mouse genome. NEUROD1 motifs were identified in mm10 mouse genome by HOMER (ver. 4.10)[139] using internal NeuroD1(bHLH)/Islet-NeuroD1-ChIP-Seq (GSE30298) motif file and converted to bedfile. Data were visualized in IGV (ver. 2.12.3)[193] and final images were generated with UCSC browser[194,195] as mean values.

## Single-cell RNA sequencing and analysis

We used adult mice *Neurod1CKO* (genotype: *Neurod1loxP/loxP;Isl1Cre/+*) and *Control* (*Neurod1loxP/+* or *Neurod1loxP/loxP* genotype from the same breeding) to isolate pancreatic islets. Mouse pancreatic islets were isolated (and pooled) from four mice per each genotype by perfusion of the common bile duct with Collagenase P (1 mg/ml; Merck) in HBS (Merck) supplemented with Actinomycin D (25 μg/ ml; Merck) to stop transcription, digestion of the pancreases with 0.8 mM Collagenase P and purification of the islets by Ficoll discontinuous gradient. Single cell dispersion was carried out as follows: islets were settled by centrifugation (140*g*, 4 °C, 4 min), washed with 2 ml of HBS with Actinomycin D (2.5 μg/ ml). Islets were centrifuged (140*g*, 4 °C, 4 min) again and Dulbecco's PBS was used for the final wash. Digestion was performed with 0.6 ml of trypsin/0.53 mM EDTA[179] supplemented with Actinomycin D (25 μg/ ml). Samples were incubated at 37 °C for 5 min. Following incubation, trypsinization was stopped by adding of 1.2 ml

of PBS with 10 mM EGTA supplemented with 10% fetal bovine serum (FBS; Merck). Cells were gently dispersed mechanically using a P1000 pipette. Cells were then spun down (800 × g, 4 °C, 10 min) and resuspended in PBS with 10 mM EGTA supplemented with 2% FBS. Single-cell suspension was used for library preparation: Chromium Next GEM Single Cell 3' Reagent Kits v3.1 (10 × Genomics, Pleasanton, CA) was used to prepare the sequencing libraries, according to the manufacturer's protocols. Briefly, 10 × Chromium platform was used to encapsulate individual cells into droplets along with beads covered in cell-specific 10 × Barcodes, unique molecular identifiers (UMIs) and poly(dT) sequences. After reverse transcription, the cDNA libraries were amplified (13–14 cycles), fragmented and ligated to sequencing adaptors. SPRISelect magnetic beads were used for purification of the cDNA suspension and size selection of the fragments. The concentration and quality of the libraries were measured using Qubit dsDNA HS Assay Kit (Invitrogen) and Fragment Analyzer HS NGS Fragment Kit (#DNF-474, Agilent). The libraries were pooled and sequenced in paired-end mode using Illumina NovaSeq 6000 SP Reagent Kit.

Single-cell sequencing data was aligned to a custom reference mouse genome GRCm38 containing the active transcript sequence of tdTomato. Aligned reads were annotated by STARsolo (STAR version 2.7.3a)[180] to the custom GENCODE version M8 annotation containing the transcript information of tdTomato. Seurat R package (version 4.3.0)[196] was used to visualize, cluster and annotate clusters on SCTransformed and integrated data. EmptyDrops function (DropletUtils R package)[197] was applied to preserve only cell-containing droplets. DoubletFinder[198] R package was used for the identification of doublets. Seurat functions FindMarkers and DoHeatmap were used for differential expression analysis and heatmaps production respectively. Proportions of cell types were extracted with Prop.table function.

## Experimental design, statistics and reproducibility

All comparisons were made between animals with the same genetic background, typically littermates, and we used male and female mice. The number of samples (n) for each comparison are given in the corresponding figure legends. Phenotyping and data analysis were performed blind to the genotype of the mice. All values are presented either as the mean ± standard deviation (SD) or standard error of the mean (SEM). For statistical analysis, GraphPad Prism (ver. 9.3.1) software and RStudio (ver. 2022.7.1.554) were used. To assess differences in the mean, one-way ANOVA with Tukey's multiple comparison test, and unpaired two-tailed $t$ tests were employed. A $P$ value < 0.05 denoted the presence of a statistically significant difference. Significance was determined as $P < 0.05$ (*), $P < 0.01$ (**), $P < 0.001$ (***), $P < 0.0001$ (****). The complete results of the statistical analyses are included in the figure legends. Each experiment presented was independently repeated a minimum of three times, unless otherwise specified.

## Reporting summary

Further information on research design is available in the Nature Portfolio Reporting Summary linked to this article.

## Data availability

All raw sequencing data have been deposited at NIH GEO. The bulk RNA-seq and Cut&Tag-seq data are under accession number GSE212084 and the single-cell RNA-seq data are under accession number GSE236559. Analyzed data are in Supplementary Data 1 and Data 2. Source data are provided with this paper.

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

## Acknowledgements

This research was supported by the Czech Science Foundation (GA19-07378S and GA22-11516S to GP), and by the institutional support of the Czech Academy of Sciences (RVO: 86652036 to GP). We thank Prof. Lori Sussel (University of Colorado) for reading and commenting of the revised MS, and A. Pavlinek (King's College London) for editing the MS. We thank M. Anderova for providing *tdTomato-Ai14* reporter line. We acknowledge Imaging Methods Core Facility at BIOCEV supported by the MEYS CR (Large RI Project LM2018129 Czech-BioImaging) and ERDF (project No. CZ.02.1.01/0.0/0.0/18_046/0016045) for its support with obtaining imaging data and FACS experiments presented in this paper; the Light Microscopy Core Facility of the Institute of Molecular Genetics of the Czech Academy of Sciences for its support for LFSM generated 3D images; Biocev GeneCore Facility for its support with gene expression/transcriptome analyses; and Biocev Animal facility (LM 2018126 Czech Centre for Phenogenomics by MEYS OP RDE CZ.02.1.01/0.0/0.0/18_046/0015861 CCP Infrastructure Upgrade II by MEYS and ESIF).

## Author contributions

G.P., R.B., and F.S. conceived and designed the study; R.B., V.F., O.S., and Z.B., performed experiments; R.B., and V.F. performed 3D visualization; V.F., P.A., and L.L-M performed single-cell RNA-seq analyses; S.B., L.V., O.S., and D.Z. performed bioinformatic analyses of bulk-RNA-seq and CUT&Taq-seq; O.S. wrote the first original draft; R.B., and V.F. contributed to the manuscript review & editing; G.P. wrote the manuscript and obtained research funding.

## Competing interests

The authors declare no competing interests.
