## [Peer Review File · Nature Communications]

NEUROD1 reinforces endocrine cell fate acquisition in pancreatic developmentREVIEWER COMMENTS

Reviewer #1 (Remarks to the Author):

In this article, the authors use previously established mouse lines to perform a new cross, namely Neurod1loxP/loxP (Goebbels et al., 2005) with Neurod1Cre/+ (Li et al., 2012) and additionally the Ai14-tdTomato allele that enables to track the cells that have expressed NeuroD1. While inactivation of NeuroD1 has been previously performed in the mouse pancreas during development and after birth, this new line enables inactivation in the very cells that express the gene, though one may say this happens with a little delay because the gene needs to get transcriptionally activated before there is enough Cre to inactivate it (a minor limitation in the present study). Previous deletions have been performed as a full knock-out (Naya et al., 1997; Dudek et al., 2021- with focused analysis on the pancreas) or conditionally in the whole pancreas at the onset of its development (Pdx1-Cre Mastracci et al., 2012), in the endocrine progenitors that are just before NeuroD1 expression (Neurog3-Cre Romer et al., 2019), in parallel to NeuroD1 expression (Insm1-Cre Dudek et al., 2021 and Isl1-Cre Bohuslavova et al., 2021) or in beta cells (Ins-Cre Romer et al., 2019, RIP-Cre Gu et al., 2020, Pdx-CreER Gu et al., 2010). The previous studies have established NeuroD1 as a gene whose inactivation (or mutations causing reduced activity) causes diabetes (MOD6 in human). Previous work has also revealed that its absence causes an impaired perinatal proliferative expansion of α and β cells, defective maturation of β cells as marked by extremely reduced Ins1 expression (but not ins2) and also a reduction in the expression of many other transcription factors, beta cell functional genes and metabolic regulators. Moreover an altered islet architecture was reported (see notably Gu et al., 2010). From the previous studies, it is clear that NeuroD1 is important for β cell function and also for their formation (see notably Dudek et al. 2021 and Bohuslavova et al., 2021). The present article confirms all these phenotypes and can thus be viewed as confirmatory for about 50% of the results. This is valuable and gives confidence in our understanding of NeuroD1 function. The study is very well executed, provides top-level illustrations and a clear quantification and description of the phenotypes. A few aspects are novel but do not lead to a change in paradigm such as the persistent expression of PDX1 in the differentiating α -cell lineage in Neurod1ST (line 154). The real novelty comes at line 158 and the authors do not feature this as prominently as they should. Although we knew that less endocrine cells expressing hormones were found during development and after birth upon NeuroD1 inactivation, what remains unclear is what happens to the endocrine progenitors, once they have started their endocrine differentiation program when they do not express NeuroD1? This had not been addressed because no reporter line tracing recombined cells had been used. The authors clearly establish here that these cells do not appear to die, though they should quantify the number of traced cells in the wt vs KO conditions. There may indeed be multiple effects with some cells dying and others trapped in an endocrine precursor state. This trapped endocrine precursor state is reminiscent of what has been seen in the Insm1 KO (Gierl et al., 2006).

The tracer allele also enabled the authors to isolate the descendants of the NeuroD1-positive cells and analyze their state by transcriptome profiling as well as cut and tag, the latter providing information on the epigenome. This is somewhat reminiscent of what was done by Dudek et al., 2021 with the Insm1-GFPCre though it is not clear whether the GFP-positive cells in this case also comprised undifferentiated endocrine precursors. While this tracing may in principle enable to characterize the state of undifferentiated endocrine precursors in the submitted article, unfortunately the reporter also labels cells that have started to express hormones. The population is thus largely heterogeneous and it is therefore not clear what the changes in gene expression and epigenome mean. Are the genes reduced in all cells or only the ones that do not express hormones? This limits the interest of these analyses. However, it succeeds in establishing a list of genes and processes impaired in average in NeuroD1 deficient endocrine progeny.

Taken together, the article is a mix of confirmation of previous observations using a different mouse line and an interesting new finding of cells accumulating in a partially differentiated state. The characterization of this state is of limited interest despite the great care involved in the molecular

analyses as it is the average of a heterogeneous population.

Reviewer #2 (Remarks to the Author):

In this manuscript, the authors characterize a genetic mouse model lacking NEUROD1 in the Neurod1-expressing endocrine lineage. The authors conclude the NEUROD1 reinforces endocrine cell fate during pancreas development by transcriptionally regulating key endocrine genes. I have some concerns over the article as currently written: 1) there needs to be further validation of the NeuroD1-cre transgenic line; 2) further investigation of what protein(s) endocrine cells express in the absence of NEUROD1; and 3) the final model

Major points:

- 1) I have some concerns over the NeuroD1-ST model.
 - a. The authors state that the Cre-expressing allele has no phenotype; this data is important to include in the manuscript, especially as the authors combine cre⁺ and cre⁻ for controls.
 - b. There also appears to be a discrepancy between the transgenic cre allele (tdTomato) and NEUROD1 protein expression (Fig S1A). How specific is this cre allele? The authors should perform tdTomato and NEUROD1 staining of the Neurod1ST model.
 - c. Fig. S1B: The left panel for NEUROD1 expression does not match the merge? These images don't appear to be consistent with a 65% reduction in NEUROD1 expression.
- 2) Figure 1D v 1E: the Neurod1ST model appears to have a major exocrine phenotype based on the AMY2 staining in panel D. However, the Neurod1ST-Ai14 appear to have normal acinar tissue. What is the difference between these two models? The authors claim there are defects in islet cell architecture using light sheet microscopy, but what about the exocrine tissue? Do you see tdTomato⁺ acinar cells? A defect in acinar cells would be consistent with the literature
- 3) Fig 1E: the inset panel for Neurod1ST is not correct.
- 4) Global Neurod1 knockout mice have increased apoptosis in the developing pancreas. TUNEL staining in your Neurod1ST model during late embryogenesis is important. It, along with decreased proliferation, may be contributing to the rapid onset of hyperglycemia neonatally.
- 5) Fig 2B: what's the white staining?
- 6) Fig 2C: persistent expression of PDX1 in alpha cells. I don't see much of a difference; can this be quantified?
- 7) Line 157: the deletion of Neurod1 results in dysregulated gene expression in differentiating alpha cells – which data supports this conclusion?
- 8) Line 160: a large portion of the endocrine tdTomato⁺ cells did not produce either GCG or INS hormones at E15.5. Is this in the KO cells only? What are they expressing then?
- 9) Figure 3C: why don't C-peptide 1 and INS overlap in the Neurod1ST model? Does insulin antibody detect proinsulin?
- 10) What does Sox9 protein expression look like in the Neurod1ST model? It's upregulated at the transcript level but are tdTomato⁺ cells coexpressing Sox9?
- 11) Fig. 5D: I don't understand your model as presented. In the absence of Neurod1 there are more acinar/ductal cells?

Reviewer #3 (Remarks to the Author):

This manuscript uses a clever method of Neurod1 self-termination (Neurod1ST) to ask questions about the role of Neurod1 in the genetic program of endocrine cell differentiation. Evidence are shown that elimination of Neurod1 using Neurod1ST results in severe neonatal diabetes, caused by reduced endocrine cell mass. Further analyses show reduced expression of beta cell maturation genes, and

increased expression of non-endocrine genes in the mutant cells. Transcriptomic and epigenomic analyses suggest that Neurod1 enforces an endocrine differentiation gene regulatory network during endocrine differentiation. The data shown are mostly convincing, and the conclusions are mostly supported by the results. Overall the findings are interesting and potentially important, and they add a new layer to our understanding of the genetic mechanisms of endocrine cell differentiation.

Major points:

1) Figures 1-3 show immunostaining of pancreatic tissue (either sections or whole mounts), which suppose to show reduces expression on endocrine genes/cell numbers in the Neurod1ST embryos. However, quantification is shown for only few of those, and the conclusions drawn from the staining are otherwise quite hard to see. It is suggested to add quantification to all panels in figures 1-3.

2) Figures 4 and 5 show results of bulk transcriptomic and epigenomic analyses to suggest that in the absence of Neurod1, endocrine cell differentiation is "confused", with cells expressing less of mature endocrine genes, and more non-endocrine genes. It is unclear if the cells are all "confused" the same (i.e. all have part endocrine, part non-endocrine gene expression), or if some cells are "true endocrine" and some just do not differentiate. While it is discussed that single cell RNA seq is hard in this model, and this is acceptable, the question can be resolved in other ways, such as staining WT and mutant pancreata with several endocrine and non-endocrine markers, and seeing if they co-localize in the same cells.

3) The system of Neurod1ST, while clever, has one important caveat: because Neurod1 is deleted in cells that start to express Neurod1 (required for the expression of Neurod1-cre), its efficacy relies on the half-life of the intact Neurod1 that is transcribed from the floxed allele at the same time (meaning: Neurod1 promoter expression is required to delete Neurod1 from the floxed allele, resulting in some Neurod1 protein from this allele present in the cells, which can start activating/repressing genes until the knockout is complete). This means that the results, at least at the early stages, are confounded. To overcome this, it would be necessary to show similar results when Neurod1 is deleted using an earlier Cre, such as Sox9-cre or Neurog3-cre. If these show complete absence of endocrine cells, it would suggest that the partial differentiation seen using Neurod1ST may be caused by early Neurod1 expressed before all floxed alleles are deleted.

Minor points:

1) In figure 1D, top panels ("control"), the green GCG staining appears nuclear, and in the same cells as the white INS staining.

2) Page 10, line 274 - should be 4F, not 3F.

3) Page 14, line 324 - the sentence "Thus Neurod1 deficiency" seems to be cut.

Reviewer #4 (Remarks to the Author):

NEUROD1 reinforces endocrine cell fate acquisition in pancreatic development

Summary: The Authors study the role of the transcription factor NEUROD1 in mammalian endocrine cell differentiation. NEUROD1 is known to reprogram neural cell types. Several lines of evidence point to an important role for this factor in endocrine differentiation programs as well. The Authors generate a conditional knockout mouse model called Neurod1ST that targets Neurod1 in early endocrine cell clusters. They first qualitatively describe the effects of Neurod1ST on pancreas formation at the tissue and organ level to characterize the physiological consequences of Neurod1 knockdown. They then

perform transcriptomic and epigenetic assays to assess how Neurod1 knockdown alters cell fate decisions in endocrine cells. Overall, the work provides a novel study of a key differentiation factor in a cell type that hasn't been shown before but lacks clarity and connection among the various results into a cohesive narrative.

Major Comments:

-First, I found the Introduction and Discussion insufficient to provide enough context about what is known about Neurod1 in other cell types (neuronal), and how the new study in endocrine cells informs our understanding of Neurod1 function in general. It is cited (e.g. lines 63 and 64) that NEUROD1 is involved in neuronal differentiation as a pioneer factor. I would like to know what has already been reported regarding remodeling of the epigenetic landscape in neuronal cells, and whether what the Authors report in endocrine cells reflects a similar or divergent function of Neurod1.

-The effects endocrine tissue architecture shown in Figure 1D and E are unclearly labeled. First, a quantification of the observed reduction in endocrine tissue content would be helpful. Is Figure 1D a single replicate of the imaging experiment? How are the boundaries of the box in which there appears to be a qualitative reduction in GCG (green) and INS (white) cells chosen? In both images cells appear outside of the box that highlights the reduction. In Figure 1E, indicating where the alpha-cell mantle exists and does not exist in the images would be helpful. Line 120 is the first time the alpha-cell mantle is discussed. This section needs to clarify the impact of the measured changes in tissue structure on pancreas function or physiology.

-The Authors claim that the deviations in gene expression for endocrine and non endocrine genes support the phenotypic pathophysiology (line 297) but there is no clear consequence of upregulated non-endocrine genes on physiology of the pancreas described. The Authors also specifically cite an interesting effect on cell cycling, that is not validated in the intact tissue. What are the consequences of defective cell cycling on pancreas pathophysiology?

-In Figure 4I, the Neurod1 motif is reported to be found in 10% of downregulated genes. This text in lines 274-275 claim that there are significant enrichments for Neurod1 binding sites in the top 10 downregulated genes. Could the authors provide context for what is considered significant? 10% seems like a low number.

-The Authors find an interesting upregulation of endocrine genes concomitant with a downregulation of non-endocrine genes. This appears to point to a role for Neurod1 in repressing non endocrine fates. The study of pioneering activity of Neurod1 points to a potential mechanism but this connection is very unclear and incomplete. Line 324 appears to have the start of a significance statement that is not finished. Framing this result in context of what is already known about Neurod1 in other cell types would be helpful. Is reprogramming by repression and activation of subsets of genes by changing chromatin state (bivalency) unique to endocrine cells? Further discussion about known pioneering activity of Neurod1 seems necessary to make a claim that the effect in the pancreas is new as stated in the text.

Minor Comments:

-Line 100 Neurod1-"self-terminating" needs to be labeled as Neurod1ST here

-Line 64 reprogramming is spelled incorrectly

-Line 121 "proliferation rate" implies a dynamic measurement of cell division. Perhaps "extent of proliferation" is a better phrase

-Line 138 A reference to a figure relating to the statement that the control islets "shared a more spherical shape, regulate size span, and coremantle organization" appears to be missing

-Line 177 The phrasing “the molecular basis of Neurod1 elimination” seems inconsistent with what the section is about (transcriptomic profiling of the effects of Neurod1 knockdown)

REVIEWER COMMENTS

Reviewer #1 (Remarks to the Author):

In this article, the authors use previously established mouse lines to perform a new cross, namely *Neurod1loxP/loxP* (Goebbels et al., 2005) with *Neurod1Cre/+* (Li et al., 2012) and additionally the Ai14-tdTomato allele that enables to track the cells that have expressed NeuroD1. While inactivation of NeuroD1 has been previously performed in the mouse pancreas during development and after birth, this new line enables inactivation in the very cells that express the gene, though one may say this happens with a little delay because the gene needs to get transcriptionally activated before there is enough Cre to inactivate it (a minor limitation in the present study). Previous deletions have been performed as a full knock-out (Naya et al., 1997; Dudek et al., 2021- with focused analysis on the pancreas) or conditionally in the whole pancreas at the onset of its development (*Pdx1-Cre* Mastracci et al., 2012), in the endocrine progenitors that are just before NeuroD1 expression (*Neurog3-Cre* Romer et al., 2019), in parallel to NeuroD1 expression (*Insm1-Cre* Dudek et al., 2021 and *Isl1-Cre* Bohuslavova et al., 2021) or in beta cells (*Ins-Cre* Romer et al., 2019, *RIP-Cre* Gu et al., 2020, *Pdx-CreER* Gu et al., 2010). The previous studies have established NeuroD1 as a gene whose inactivation (or mutations causing reduced activity) causes diabetes (MOD6 in human). Previous work has also revealed that its absence causes an impaired perinatal proliferative expansion of α and β cells, defective maturation of β cells as marked by extremely reduced *Ins1* expression (but not *Ins2*) and also a reduction in the expression of many other transcription factors, beta cell functional genes and metabolic regulators. Moreover, an altered islet architecture was reported (see notably Gu et al., 2010). From the previous studies, it is clear that NeuroD1 is important for β cell function and also for their formation (see notably Dudek et al. 2021 and Bohuslavova et al., 2021). The present article confirms all these phenotypes and can thus be viewed as confirmatory for about 50% of the results. This is valuable and gives confidence in our understanding of NeuroD1 function. The study is very well executed, provides top-level illustrations and a clear quantification and description of the phenotypes.

We thank the reviewer for the overall positive assessment of our study in light of previous studies in the field.

A few aspects are novel but do not lead to a change in paradigm such as the persistent expression of PDX1 in the differentiating α -cell lineage in *Neurod1ST* (line 154). The real novelty comes at line 158 and the authors do not feature this as prominently as they should. Thank you for pointing this out. We quantified: (i) the number of cells co-expressing PDX1 and glucagon (revised Fig. 2); and (ii) the number of tdTomato positive cells that did not produce either glucagon or insulin in the Control and *Neurod1ST* pancreas at E15.5 (revised Fig. 3).

Although we knew that less endocrine cells expressing hormones were found during development and after birth upon NeuroD1 inactivation, what remains unclear is what happens to the endocrine progenitors, once they have started their endocrine differentiation program when they do not express NeuroD1? This had not been addressed because no reporter line tracing recombined cells had been used. The authors clearly establish here that these cells do not appear to die, though they should quantify the number of traced cells in the

wt vs KO conditions. There may indeed be multiple effects with some cells dying and others trapped in an endocrine precursor state. This trapped endocrine precursor state is reminiscent of what has been seen in the *Insm1* KO (Gierl et al., 2006). The tracer allele also enabled the authors to isolate the descendants of the NeuroD1-positive cells and analyze their state by transcriptome profiling as well as cut and tag, the latter providing information on the epigenome. This is somewhat reminiscent of what was done by Dudek et al., 2021 with the *Insm1*-GFPCre though it is not clear whether the GFP-positive cells in this case also comprised undifferentiated endocrine precursors. While this tracing may in principle enable to characterize the state of undifferentiated endocrine precursors in the submitted article, unfortunately the reporter also labels cells that have started to express hormones. The population is thus largely heterogeneous and it is therefore not clear what the changes in gene expression and epigenome mean. Are the genes reduced in all cells or only the ones that do not express hormones? This limits the interest of these analyses. However, it succeeds in establishing a list of genes and processes impaired in average in NeuroD1 deficient endocrine progeny.

Taken together, the article is a mix of confirmation of previous observations using a different mouse line and an interesting new finding of cells accumulating in a partially differentiated state. The characterization of this state is of limited interest despite the great care involved in the molecular analyses as it is the average of a heterogeneous population.

Thank you for your valuable feedback. We appreciate your positive comments regarding our confirmation of previous observations and the identification of an interesting new finding in our study. We understand your concerns regarding the limited interest in the characterization of partially differentiated cells due to the heterogeneity of the population.

In response to your suggestion, we have performed single-cell RNA sequencing to further investigate the endocrine population in the conditional *Neurod1* deletion mouse line. By analyzing the transcriptomes of individual cells, we aimed to identify subpopulations of endocrine cells and determine whether the partially differentiated state we observed is a distinct population or a transitional state. This analysis provides a more detailed understanding of the effects of *Neurod1* deletion on endocrine cell differentiation (new Fig. 6 and 7, Supplementary Fig. 5).

Thank you again for your helpful comments. We hope that our revised approach addresses your concerns and improves the quality and impact of our study.

Reviewer #2 (Remarks to the Author):

In this manuscript, the authors characterize a genetic mouse model lacking NEUROD1 in the Neurod1-expressing endocrine lineage. The authors conclude the NEUROD1 reinforces endocrine cell fate during pancreas development by transcriptionally regulating key endocrine genes. I have some concerns over the article as currently written: 1) there needs to be further validation of the NeuroD1-cre transgenic line; 2) further investigation of what protein(s) endocrine cells express in the absence of NEUROD1; and 3) the final model

Major points:

1) I have some concerns over the NeuroD1-ST model.

a. The authors state that the Cre-expressing allele has no phenotype; this data is important to include in the manuscript, especially as the authors combine cre⁺ and cre⁻ for controls.

Thank you for your valuable feedback. We agree that it is important to include data demonstrating that the Cre-expressing allele has no phenotype in our manuscript, particularly as we used both Cre⁺ and Cre⁻ mice as controls.

In response to your suggestion, we have included an additional summary in the Results section of the revised manuscript (page 5) pointing out the results of the phenotypes of Cre⁺ (heterozygous *Neurod1*Cre⁺; *Neurod1*^{loxP/+}) and Cre⁻ “Control” (*Neurod1*-Cre^{negative}; *Neurod1*^{loxP/loxP} or *Neurod1*Cre^{negative}; *Neurod1*^{loxP/+}) mice. These mice had no discernible phenotype, confirming that the effects we observed in the *Neurod1*-eliminated mouse line (genotype *Neurod1*Cre⁺; *Neurod1*^{loxP/loxP}) are specific to the deletion of *Neurod1* and not due to any unintended effects of the Cre recombinase. These results showed that there are no differences in blood glucose levels, body weight, survival, and the formation of endocrine islets of Langerhans in between Cre⁺ and Cre^{negative} mice (Fig. 1, Fig. 2, Supplementary Fig. 2).

b. There also appears to be a discrepancy between the transgenic cre allele (tdTomato) and NEUROD1 protein expression (Fig S1A). How specific is this cre allele? The authors should perform tdTomato and NEUROD1 staining of the NeuroD1ST model.

To address the reviewer's concern, we performed additional immunohistochemical analyses and included new supplementary images in our revised manuscript (new Supplementary Fig. 1). We included images showing that the tdTomato reporter is co-expressed with glucagon in the first differentiated endocrine cell clusters of the pancreas at E10.5 (Supplementary Fig. 1a). Furthermore, we added new images showing tdTomato expression domain with co-expression NEUROD1 in *Control-Ai14*, and new images of the *NeuroD1ST* pancreas with a significant reduction of NEUROD1 expression in tdTomato positive endocrine clusters (Supplementary Fig. 1b, c). These new analyses provide strong evidence for the specificity of the Neurod1^{Cre}, as tdTomato expression is exclusively detected within the endocrine cell clusters.

c. Fig. S1B: The left panel for NEUROD1 expression does not match the merge? These images don't appear to be consistent with a 65% reduction in NEUROD1 expression.

We performed new immunohistochemical analyses using new anti-NEUROD1 antibody. The panels with new images clearly show the reduction in NEUROD1 expression, consistent with a 65% reduction in NEUROD1 levels (Supplementary Fig. 1b, c).

2) Figure 1D v 1E: the NeuroD1ST model appears to have a major exocrine phenotype based on the AMY2 staining in panel D. However, the NeuroD1ST-Ai14 appear to have normal acinar tissue. What is the difference between these two models? The authors claim there are defects in islet cell architecture using light sheet microscopy, but what about the exocrine tissue? Do you see tdTomato+ acinar cells? A defect in acinar cells would be consistent with the literature.

Thank you for bringing this to our attention. We have carefully reviewed our data and conducted additional analysis, including examining additional sections with α -amylase staining.

We acknowledge that the previous image included in Fig. 1D only showed the periphery of the acinar pancreas in *Neurod1ST*, which could have led to potential misinterpretation. Therefore, we have replaced the images to show acinar cells in *Neurod1ST*, and we have used the same colors for α -amylase staining to ensure consistency and avoid any confusion in the revised Fig. 1d and 1e. We did not observe tdTomato expression in acinar cells. Importantly, to our knowledge, there have been no previous studies reporting defects specifically in acinar cells in *Neurod1* deletion models. The *Neurod1* deletion models have been extensively studied in the context of endocrine cell development, and previous investigations have consistently reported on the effects of *Neurod1* deletion on endocrine cell populations. However, the impact of *Neurod1* deletion on acinar cells has not been a focus of these studies.

3) Fig 1E: the inset panel for Neurod1ST is not correct.

Thank you for your feedback regarding Fig. 1E. We apologize for any confusion caused by the discrepancy in the appearance of the *Neurod1ST* inset panel and the larger magnification panel. To address your concern, we have carefully examined the original image data and found that the discrepancy in the appearance of the image is due to the different number of Z-stacks used for the larger magnification. As a result, the image appears different despite being taken from the same area. To address this issue and avoid any confusion, we have replaced the image with a new one, so both the inset and larger magnification panels show a similar view. This new image will provide a clearer and more consistent representation of the labeled cells.

4) Global *Neurod1* knockout mice have increased apoptosis in the developing pancreas. TUNEL staining in your Neurod1ST model during late embryogenesis is important. It, along with decreased proliferation, may be contributing to the rapid onset of hyperglycemia neonatally.

Consistently all studies including ours demonstrate that *Neurod1* elimination prior pancreatic endocrine cell differentiation reduced endocrine cells at birth resulting in severe neonatal diabetes. However, recently it has been proposed that the major cause of endocrine cell deficient numbers is primarily proliferation defects and not apoptosis (Bohuslavova et al., 2021; Romer et al., 2019). These differences may have been previously overlooked because of the limited markers and less sensitive assays in the first study (Naya et al., 1997). To address the reviewer's concern, we performed TUNEL analyses (Supplementary Fig. 3), which confirmed recent studies findings that apoptosis has only a minor contribution to a reduced number of endocrine cells in the *Neurod1* mutant pancreas.

The first study investigating the *Neurod1* deficient pancreas by Naya *et al.* (Naya et al., 1997) demonstrated a small increase of apoptosis by only $\sim 1.5\%$ in X-gal-stained (*Neurod1-lacZ* fusion protein) E17.5 pancreatic sections of *Neurod1* KO. The level of apoptosis cannot explain a massive reduction ($\sim 60\%$) of β -gal-expressing endocrine cells in *Neurod1* KO at E17.5. Additional TUNEL analyses of P0 pancreatic sections demonstrated $\sim 10\%$ apoptotic X-Gal-stained endocrine cells. However, these results might be negatively affected by severe neonatal diabetes and limited survival of *Neurod1* KO. This conclusion is further supported

by the fact that the percentage of endocrine cells is similar between E17.5 and P0, representing 57% and 59% reduction in *Neurod1* KO compared to control pancreases (Naya et al., 1997). Recent studies demonstrated that the elimination of *Neurod1* prominently affects proliferation and apoptosis has only a minor contribution to a lower number of pancreatic endocrine cells. For example, Romer et al. (Romer et al., 2019) showed that apoptotic TUNEL⁺ insulin-expressing cells at E18 were extremely rare, representing ~ 0.25% of insulin⁺ cells in *Neurod1* CKO pancreas. They did not detect any measurable apoptosis in earlier developmental points (E15 and E17). Similarly, Bohuslavova et al. (Bohuslavova et al., 2021) showed a minor not significant increase of apoptotic insulin⁺ and glucagon⁺ cells in the pancreas of *Neurod1* CKO mutant at E17.5, in contrast to significant proliferation deficit in endocrine cells lacking *Neurod1*. Additional study by Dudek et al. (Dudek et al., 2021) confirmed significant reduced proliferation of GFP positive endocrine cells in embryos lacking *Neurod1* (2.8% vs 15% in control cells).

In summary, our TUNEL analyses support the recent conclusions that the elimination of *Neurod1* primarily affects the proliferation of pancreatic endocrine cells, while apoptosis only has a minor contribution to the decreased number of endocrine cells.

5) Fig 2B: what's the white staining?

We have carefully checked the image, there is no white staining, it is green (see below, the image split to the respective individual colors).

6) Fig 2C: persistent expression of PDX1 in alpha cells. I don't see much of a difference; can this be quantified?

Thank you for pointing this out. We quantified number of co-expressing PDX1⁺ and GCG⁺ cells (revised Fig. 2c).

7) Line 157: the deletion of *Neurod1* results in dysregulated gene expression in differentiating alpha cells – which data supports this conclusion?

Thank you for your comment. We appreciate the opportunity to provide more context to support our conclusion. This was based on the persistent expression of PDX1 in glucagon expressing cells in *Neurod1ST*. We revised the sentence (line 173): “Although the generation of GCG⁺ endocrine cells was not affected, the persistent expression of PDX1 in GCG⁺ cells, indicates that the differentiation of these cells might be altered or delayed in the absence of NEUROD1.”

8) Line 160: a large portion of the endocrine tdTomato⁺ cells did not produce either GCG or INS hormones at E15.5. Is this in the KO cells only? What are they expressing then?

We agree that this is an important point. The secondary transition wave from E12.5 to E16.5 is characterized by a massive differentiation of pancreatic lineages from delaminating epithelial cells. In our study, we observed that a significant proportion of the endocrine tdTomato⁺ cells at E15.5 did not express either GCG or INS hormones (quantified in the revised Fig. 3). A significantly higher frequency of these cells is observed in the *Neurod1*ST mutant pancreas. It suggests that these tdTomato⁺ cells might have an altered or disrupted differentiation process.

Our transcriptomic analyses provide further evidence supporting the notion that endocrine cell differentiation is impaired in the *Neurod1*ST mutant pancreas. This could explain the observed discrepancy between tdTomato labeling and hormone expression in these cells. The exact identity and molecular characteristics of these non-hormone-expressing tdTomato⁺ cells require further investigation.

9) Figure 3C: why don't C-peptide 1 and INS overlap in the Neurod1ST model? Does insulin antibody detect proinsulin?

Upon reevaluating our analyses and reviewing the data, we confirm that there is indeed overlap between C-peptide 1 and INS in the *Neurod1*ST model.

10) What does Sox9 protein expression look like in the Neurod1ST model? It's upregulated at the transcript level but are tdTomato⁺ cells coexpressing Sox9?

We agree that this is an important avenue to explore. Based on the RNAseq of E15.5 endocrine cells, we found that the transcript level expression of *Sox9* was significantly upregulated in the *Neurod1*ST compared to control, as mentioned in the manuscript. Therefore, we immunolabeled the sections of pancreas with anti-SOX9, -PDX1, and -insulin. Our protein-level data confirmed the co-expression of SOX9 and tdTomato (revised Fig. 4). We believe that our provides valuable insight into the potential role of NEUROD1 in regulating Sox9 expression during pancreatic development.

11) Fig. 5D: I don't understand your model as presented. In the absence of Neurod1 there are more acinar/ductal cells?

Thank you for your comment. Using RNA-seq and CUT&Tag-seq, we analyzed only endocrine cells expressing tdTomato. Our RNA-seq analysis of *Neurod1*ST endocrine cells revealed dysregulation of the transcriptome, characterized by reduced expression of genes associated with endocrine function and enrichment of non-endocrine genes related to the exocrine system and pancreatic ductal cells. These data indicate that the differentiation of endocrine cells with *Neurod1* deletion is altered. This ambiguous cell fate commitment of endocrine cells lacking *Neurod1* was associated with functional and morphological defects of the endocrine pancreas, including a decrease in the number of endocrine cells, reduced insulin production, and neonatal diabetes. Thus, our results indicate that NEUROD1 is important for both activation of endocrine cell type-specific genes and repression of genes associated with alternative non-endocrine lineages.

We apologize for any confusion in our previous explanation and have updated the figure legend to provide a more comprehensive explanation of our findings (see Fig. 8 legend). We hope that this clarifies the conclusions of our study and thank you again for your comment and feedback.

Reviewer #3 (Remarks to the Author):

This manuscript uses a clever method of Neurod1 self-termination (Neurod1ST) to ask questions about the role of Neurod1 in the genetic program of endocrine cell differentiation. Evidence are shown that elimination of Neurod1 using Neurod1ST results in severe neonatal diabetes, caused by reduced endocrine cell mass. Further analyses show reduced expression of beta cell maturation genes, and increased expression of non-endocrine genes in the mutant cells. Transcriptomic and epigenomic analyses suggest that Neurod1 enforces an endocrine differentiation gene regulatory network during endocrine differentiation. The data shown are mostly convincing, and the conclusions are mostly supported by the results. Overall, the findings are interesting and potentially important, and they add a new layer to our understanding of the genetic mechanisms of endocrine cell differentiation.

Thank you for your positive evaluation of our manuscript and for your stimulating comments. We appreciate your feedback and are glad to hear that the findings are interesting and potentially important. We have carefully considered your suggestions and addressed any concerns or limitations that you pointed out in our revised manuscript. We hope that these changes have further strengthened our work and we appreciate your valuable input.

Major points:

1) Figures 1-3 show immunostaining of pancreatic tissue (either sections or whole mounts), which supposed to show reduces expression on endocrine genes/cell numbers in the Neurod1ST embryos. However, quantification is shown for only few of those, and the conclusions drawn from the staining are otherwise quite hard to see. It is suggested to add quantification to all panels in figures 1-3.

We added the missing quantifications (revised Figure 1-3).

2) Figures 4 and 5 show results of bulk transcriptomic and epigenomic analyses to suggest that in the absence of Neurod1, endocrine cell differentiation is "confused", with cells expressing less of mature endocrine genes, and more non-endocrine genes. It is unclear if the cells are all "confused" the same (i.e. all have part endocrine, part non-endocrine gene expression), or if some cells are "true endocrine" and some just do not differentiate. While it is discussed that single cell RNA seq is hard in this model, and this is acceptable, the question can be resolved in other ways, such as staining WT and mutant pancreata with several endocrine and non-endocrine markers, and seeing if they co-localize in the same cells.

We have taken your feedback into consideration and have made significant improvements to our study. To address the important point you raised, we conducted immunolabeling of SOX9 and have included new images and quantifications that show a marked increase in co-expression of SOX9 and tdTomato in the mutant E15.5 pancreas (please refer to revised Fig. 4).

Furthermore, we have now included new single cell RNA sequencing data that demonstrate changes in the alpha, beta, and endocrine progenitor populations as a result of *Neurod1* deletion, in comparison to the control endocrine cell populations (please refer to new Fig. 6 and 7, and new Supplementary Fig. 5). We believe that these additional results provide valuable insights into the mechanisms underlying the observed phenotypic changes in our mutant model.

Thank you for your helpful comments, which have enabled us to strengthen our study and improve the quality of our findings.

3) The system of *Neurod1*ST, while clever, has one important caveat: because *Neurod1* is deleted in cells that start to express *Neurod1* (required for the expression of *Neurod1*-cre), its efficacy relies on the half-life of the intact *Neurod1* that is transcribed from the floxed allele at the same time (meaning: *Neurod1* promoter expression is required to delete *Neurod1* from the floxed allele, resulting in some *Neurod1* protein from this allele present in the cells, which can start activating/repressing genes until the knockout is complete). This means that the results, at least at the early stages, are confounded. To overcome this, it would be necessary to show similar results when *Neurod1* is deleted using an earlier Cre, such as *Sox9*-cre or *Neurog3*-cre. If these show complete absence of endocrine cells, it would suggest that the partial differentiation seen using *Neurod1*ST may be caused by early *Neurod1* expressed before all floxed alleles are deleted.

We appreciate your concern regarding the potential confounding effect of the delayed *Neurod1* deletion in our *Neurod1*ST model. However, we would like to emphasize that our reporter analyses provide evidence for the efficient activity of *Neurod1*^{Cre} during early pancreas development. We demonstrated robust expression of the reporter tdTomato in early endocrine cell clusters and the effective deletion of *NEUROD1* protein in the developing pancreas as early as E10.5 (new Supplementary Fig. 1).

While it is true that there might be a delay in the complete knockout of *Neurod1* due to the requirement of *Neurod1* promoter expression for *Neurod1* deletion, we want to highlight that this expected delay does not undermine the overall phenotype observed in our *Neurod1*ST model. The phenotype we observed is consistent with the phenotype observed in the global *Neurod1* deletion mutant (Naya et al., 1997). In the global knockout, there is also a severe neonatal phenotype with a reduction in endocrine cell population but not a complete absence of endocrine cells.

We also want to point out that previous studies have used an earlier Cre, *Neurog3*^{Cre}, to delete *Neurod1*, and the resulting phenotype was similar to that observed in our study using the *Neurod1*ST system (Romer et al., 2019). In fact, the study by Romer et al. (2019) reported a severe neonatal diabetes phenotype, reduced number of generated alpha and beta cells, decreased proliferation of endocrine cells, and a significant defect in insulin production, particularly *Ins1*, all of which are consistent with our findings using the *Neurod1*ST system.

Therefore, despite the anticipated delay in *Neurod1* deletion, we believe that the partial endocrine differentiation observed in our *Neurod1*ST system is a common feature of both the global *Neurod1* deletion mutant and the earlier *Cre-Neurod1* conditional deletion mutants. Thus, it is a developmental feature of *Neurod1* deletion mutations, and we are confident in the validity of our results.

Minor points:

1) In figure 1D, top panels ("control"), the green GCG staining appears nuclear, and in the same cells as the white INS staining.

In the top panels ("Control") of Figure 1d, the green GCG staining appears to be nuclear and co-localized with the white INS staining in the same cells. However, it should be noted that this is not due to actual nuclear localization of GCG, but rather a result of the imaging technique used, which involves capturing Z stacks.

2) Page 10, line 274 - should be 4F, not 3F.

Thank you. It was corrected.

3) Page 14, line 324 - the sentence "Thus Neurod1 deficiency" seems to be cut.

It was removed.

Reviewer #4 (Remarks to the Author):

NEUROD1 reinforces endocrine cell fate acquisition in pancreatic development

Summary: The Authors study the role of the transcription factor NEUROD1 in mammalian endocrine cell differentiation. NEUROD1 is known to reprogram neural cell types. Several lines of evidence point to an important role for this factor in endocrine differentiation programs as well. The Authors generate a conditional knockout mouse model called Neurod1ST that targets Neurod1 in early endocrine cell clusters. They first qualitatively describe the effects of Neurod1ST on pancreas formation at the tissue and organ level to characterize the physiological consequences of Neurod1 knockdown. They then perform transcriptomic and epigenetic assays to assess how Neurod1 knockdown alters cell fate decisions in endocrine cells. Overall, the work provides a novel study of a key differentiation factor in a cell type that hasn't been shown before but lacks clarity and connection among the various results into a cohesive narrative.

Major Comments:

1) First, I found the Introduction and Discussion insufficient to provide enough context about what is known about Neurod1 in other cell types (neuronal), and how the new study in endocrine cells informs our understanding of Neurod1 function in general. It is cited (e.g. lines 63 and 64) that NEUROD1 is involved in neuronal differentiation as a pioneer factor. I would like to know what has already been reported regarding remodeling of the epigenetic landscape in neuronal cells, and whether what the Authors report in endocrine cells reflects a similar or divergent function of Neurod1.

Thank you for your comment. We agree that the Introduction and Discussion can be improved to provide more context about NEUROD1 function in other cell types, particularly neuronal cells, and how our study in endocrine cells extend our understanding of NEUROD1 function in general.

To address your concern, we have revised the Introduction to include additional information about the function of NEUROD1 in neuronal differentiation, as well as its role in epigenetic

remodeling in neurons. We also discussed the similarities and differences between NEUROD1's roles in neuronal cells and pancreatic endocrine cells (page 3, 23).

We hope that these additions will provide a more comprehensive understanding of NEUROD1 function in different cell types and help readers to better appreciate the significance of our study in endocrine cells.

2) The effects endocrine tissue architecture shown in Figure 1D and E are unclearly labeled. First, a quantification of the observed reduction in endocrine tissue content would be helpful. Is Figure 1D a single replicate of the imaging experiment? How are the boundaries of the box in which there appears to be a qualitative reduction in GCG (green) and INS (white) cells chosen? In both images cells appear outside of the box that highlights the reduction. In Figure 1E, indicating where the alpha-cell mantle exists and does not exist in the images would be helpful. Line 120 is the first time the alpha-cell mantle is discussed. This section needs to clarify the impact of the measured changes in tissue structure on pancreas function or physiology.

We appreciate the reviewer's feedback and concerns regarding the clarity of labeling and the impact of measured changes in tissue structure on pancreas function or physiology.

To address these concerns, we have made several improvements.

1. Quantification of endocrine tissue content: We have included a new quantification of endocrine cells at P0 to provide a more quantitative assessment of the observed reduction in endocrine tissue content. This will help provide a clearer understanding of the extent of the reduction.
2. Figure 1D is not a single replicate of the imaging experiment. All our immunohistochemistry analyses included in the study are based on a minimum of three distinct biological replicates.
3. Clarification of labeling and boundaries: We apologize for the confusion caused by the discrepancy in labeling and the chosen boundaries in (revised Figure 1d, e). We have carefully examined the original image data and made revisions accordingly. We have now explained the alpha-cell mantle in Figure 1 legend and an additional explanation in the Introduction (page 1). This will enhance the understanding of the alpha-cell distribution within the pancreas and the islet architecture.
4. Impact on pancreas function and physiology: We acknowledge the need to clarify the impact of the measured changes in tissue structure on pancreas function or physiology. We have extended the description of the pancreatic islet architecture in the figure legend to provide more context and explanation. Additionally, we have included a note for the alpha-cell mantle in the Introduction.

To address the concerns raised by the reviewer and to improve the clarity and accuracy of our findings, we believe these modifications will enhance the interpretation and understanding of the presented data. Thank you for bringing these points to our attention, and we will ensure that these revisions are included in the revised manuscript.

3) The Authors claim that the deviations in gene expression for endocrine and non-endocrine genes support the phenotypic pathophysiology (line 297) but there is no clear consequence of upregulated non-endocrine genes on physiology of the pancreas described. The Authors also specifically cite an interested effect on cell cycling, that is not validate in the intact tissue.

What are the consequence of defective cell cycling on pancreas pathophysiology?

In our study, we observed significant deviations in gene expression between *Neurod1ST* and control endocrine cells, including both endocrine and non-endocrine genes. While we did not directly investigate the consequence of upregulated non-endocrine genes on pancreas physiology, the enrichment of non-endocrine genes in *Neurod1ST* endocrine cells implies a defect in their differentiation process and potential alterations in cellular functions. This is further supported by the severe diabetic phenotype observed in *Neurod1ST* mice, characterized by reduced production of insulin and elevated blood glucose level (Fig. 1).

Regarding the effect on cell cycling, our RNA-seq analysis of *Neurod1ST* endocrine cells revealed significant changes in the expression of genes associated with the cell cycle when compared to *Control* endocrine cells. Defective cell cycling can disrupt the balance between cell proliferation and differentiation. As the endocrine precursors exit the self-replicating state, they start to differentiate. Our results in the *Neurod1ST* suggest that their disrupted developmental progression keeps the cell fate hanging at the differentiation crossroads, manifesting in several aspects. We show that one of them is dysregulated cell cycle (overlapping with the programmed cell death GO cohort), as the *Neurod1ST* maintains the expression of exocrine markers that are directly associated with cell cycle regulation and often with neoplastic phenotypes in adults. Previous studies have demonstrated that cell cycle regulation within endocrine precursors is tightly linked to transcription regulatory aspects of their differentiation, such as *Neurog3* (Azzarelli et al., 2017) and *Pdx1* (Zhu et al., 2021). Moreover, imbalanced proliferation-regulating factors lead to malfunctions in differentiated insulin-producing beta-cells. In most cases, upregulated proliferation markers concurrently occur with decreased beta-cell mass (Zhu et al., 2021), which would eventually cause deficiency in insulin production. Although we did not validate this effect in intact tissue, our study did demonstrate a substantial reduction in the proliferation of *Neurod1ST* endocrine cells, as indicated in Fig. 1. This finding aligns with the concept of defective cell cycling and further supports the notion that impaired proliferation may contribute to the observed phenotypic alterations. The diminished proliferation of *Neurod1ST* endocrine cells is closely associated with a decrease in the total number of alpha and beta endocrine cells (Fig. 1), highlighting the impact of defective cell cycling on endocrine cell populations.

4) In Figure 4I, the *Neurod1* motif is reported to be found in 10% of downregulated genes. This text in lines 274-275 claim that there are significant enrichments for *Neurod1* binding sites in the top 10 downregulated genes. Could the authors provide context for what is considered significant? 10% seems like a low number.

We appreciate the reviewer's comment and would like to provide further context regarding the significance of the reported enrichment for the *NEUROD1* motif among the top 10 enriched transcription factor binding motifs in downregulated genes in revised Figure 4j.

The analysis, we performed, involved searching for transcription factor binding motifs within the promoters of differentially expressed genes identified in our RNA-seq analysis of *Neurod1ST* endocrine cells at E15.5. It is important to note that this set of genes was not derived from *NEUROD1* binding ChIP-seq data with known *NEUROD1* binding motifs. To assess the significance of motif enrichment, we utilized the HOMER *de novo* motif discovery algorithm, which scores motifs based on their enrichment in the target set relative to the background set. The background set in HOMER is composed of randomly selected genomic sequences or control sequences that are matched for relevant characteristics such as GC

content, sequence length, and genomic distribution. In *de novo* motif analysis, the goal is to identify enriched motifs in a given set of DNA sequences without prior knowledge of specific motifs. The target set in our analysis consisted of genes that showed differential expression at E15.5, including both direct targets of NEUROD1 and secondary targets that may not have NEUROD1 binding sites.

In this context, the HOMER analysis identified significant enrichment for binding sites of NEUROD1 within the extracted top ten hits for the promoters of the downregulated genes. While the 10% figure may seem low, it represents a statistically significant enrichment of the NEUROD1 motif within the analyzed gene set. This finding supports our conclusion that the deletion of *Neurod1* led to the downregulation of NEUROD1 target genes. We hope this explanation provides a clearer understanding of the significance of the reported enrichment.

5) The Authors find an interesting upregulation of endocrine genes concomitant with a downregulation of non-endocrine genes. This appears to point to a role for Neurod1 in repressing non endocrine fates. The study of pioneering activity of Neurod1 points to a potential mechanism but this connection is very unclear and incomplete. Line 324 appears to have the start of a significance statement that is not finished. Framing this result in context of what is already known about Neurod1 in other cell types would be helpful. Is reprogramming by repression and activation of subsets of genes by changing chromatin state (bivalency) unique to endocrine cells? Further discussion about known pioneering activity of Neurod1 seems necessary to make a claim that the effect in the pancreas is new as stated in the text.

We appreciate the reviewer's comment and agree that further discussion is necessary to provide a clearer context for the findings regarding NEUROD1's role in repressing non-endocrine fates and its potential pioneering activity.

NEUROD1 has been previously recognized as a key transcription factor involved in cell fate determination and differentiation in neurons (Filova et al., 2022; Gao et al., 2009; Hevner et al., 2006; Miyata et al., 1999). NEUROD1 remodels the epigenetic and transcriptional landscape during neuronal differentiation or reprogramming (Akol et al., 2023; Matsuda et al., 2019; Pataskar et al., 2016). It acts as a pioneer factor by initiating and driving the transcriptional program that leads to the establishment and maintenance of specific cell identities. This pioneering activity involves both activation of cell type-specific genes and repression of genes associated with alternative non-neuronal lineages (Matsuda et al., 2019).

In our study, we observed that upon *Neurod1* deletion, there was an upregulation of non-endocrine genes in pancreatic endocrine cells, suggesting that NEUROD1 plays a role in repressing non-endocrine fates within the endocrine pancreas. The exact mechanism by which NEUROD1 achieves this repression is not fully understood but likely involves changes in chromatin states and the establishment of bivalency, characterized by the presence of H3K4me3 and H3K27me3 histone modifications. This phenomenon of bivalency has been observed in various cell types during the determination of cell fate.

We acknowledge that the connection between NEUROD1's pioneering activity and its specific role in the pancreas is still not fully elucidated and warrants further investigation. While NEUROD1's pioneering activity has been extensively studied in neurons, its role in the pancreas and its impact on endocrine cell fate specification have not been thoroughly characterized. However, our study provides an initial insight into the involvement of

NEUROD1 in the establishment of the endocrine lineage, highlighting the changes in epigenetic and transcriptional landscapes that promote endocrine cell fate.

To provide a more comprehensive understanding of our findings, we have expanded the discussion to include a broader context of NEUROD1's role in cell fate determination and differentiation, specifically focusing on its known functions in neurons. We emphasize the similarities between NEUROD1's roles in neuronal differentiation and cell fate determination and its potential role in pancreatic endocrine cell development. Through this comparison, we aim to shed light on NEUROD1's broader role as an epigenetic and transcriptional regulator in embryonic development.

Thank you for raising these points, and we have incorporated these explanations and discussions into the revised manuscript to provide a clearer and more comprehensive understanding of our findings.

Minor Comments:

- Line 100 Neurod1-“self-terminating” needs to be labeled as Neurod1ST here
- Line 64 reprogramming is spelled incorrectly
- Line 121 “proliferation rate” implies a dynamic measurement of cell division. Perhaps “extent of proliferation” is a better phrase
- Line 138 A reference to a figure relating to the statement that the control islets “shared a more spherical shape, regulate size span, and coremantle organization” appears to be missing
- Line 177 The phrasing “the molecular basis of Neurod1 elimination” seems inconsistent with what the section is about (transcriptomic profiling of the effects of Neurod1 knockdown)

Thank you for bringing these corrections to our attention. The text has been revised accordingly.

References:

- Akol, I., Izzo, A., Gather, F., Strack, S., Heidrich, S., D, O.h., Villarreal, A., Hacker, C., Rauleac, T., Bella, C., *et al.* (2023). Multimodal epigenetic changes and altered NEUROD1 chromatin binding in the mouse hippocampus underlie FOXG1 syndrome. *Proc Natl Acad Sci U S A* *120*, e2122467120.
- Azzarelli, R., Hurley, C., Sznurkowska, M.K., Rulands, S., Hardwick, L., Gamper, I., Ali, F., McCracken, L., Hindley, C., McDuff, F., *et al.* (2017). Multi-site Neurogenin3 Phosphorylation Controls Pancreatic Endocrine Differentiation. *Dev Cell* *41*, 274-286 e275.
- Bohuslavova, R., Smolik, O., Malfatti, J., Berkova, Z., Novakova, Z., Saudek, F., and Pavlinkova, G. (2021). NEUROD1 Is Required for the Early α and β Endocrine Differentiation in the Pancreas. *International Journal of Molecular Sciences* *22*, 6713.
- Dudek, K.D., Osipovich, A.B., Cartailier, J.-P., Gu, G., and Magnuson, M.A. (2021). *Insm1*, *Neurod1*, and *Pax6* promote murine pancreatic endocrine cell development through overlapping yet distinct RNA transcription and splicing programs. *G3 Genes|Genomes|Genetics* *11*.
- Filova, I., Bohuslavova, R., Tavakoli, M., Yamoah, E.N., Fritzsich, B., and Pavlinkova, G. (2022). Early Deletion of *Neurod1* Alters Neuronal Lineage Potential and Diminishes Neurogenesis in the Inner Ear. *Front Cell Dev Biol* *10*, 845461.

Gao, Z., Ure, K., Ables, J.L., Lagace, D.C., Nave, K.A., Goebbels, S., Eisch, A.J., and Hsieh, J. (2009). Neurod1 is essential for the survival and maturation of adult-born neurons. *Nat Neurosci* *12*, 1090-1092.

Hevner, R.F., Hodge, R.D., Daza, R.A., and Englund, C. (2006). Transcription factors in glutamatergic neurogenesis: conserved programs in neocortex, cerebellum, and adult hippocampus. *Neurosci Res* *55*, 223-233.

Matsuda, T., Irie, T., Katsurabayashi, S., Hayashi, Y., Nagai, T., Hamazaki, N., Adefuin, A.M.D., Miura, F., Ito, T., Kimura, H., *et al.* (2019). Pioneer Factor NeuroD1 Rearranges Transcriptional and Epigenetic Profiles to Execute Microglia-Neuron Conversion. *Neuron* *101*, 472-485.e477.

Miyata, T., Maeda, T., and Lee, J.E. (1999). NeuroD is required for differentiation of the granule cells in the cerebellum and hippocampus. *Genes Dev* *13*, 1647-1652.

Naya, F.J., Huang, H.P., Qiu, Y., Mutoh, H., DeMayo, F.J., Leiter, A.B., and Tsai, M.J. (1997). Diabetes, defective pancreatic morphogenesis, and abnormal enteroendocrine differentiation in BETA2/neuroD-deficient mice. *Genes Dev* *11*, 2323-2334.

Pataskar, A., Jung, J., Smialowski, P., Noack, F., Calegari, F., Straub, T., and Tiwari, V.K. (2016). NeuroD1 reprograms chromatin and transcription factor landscapes to induce the neuronal program. *The EMBO Journal* *35*, 24-45.

Romer, A.I., Singer, R.A., Sui, L., Egli, D., and Sussel, L. (2019). Murine Perinatal β -Cell Proliferation and the Differentiation of Human Stem Cell-Derived Insulin-Expressing Cells Require NEUROD1. *Diabetes* *68*, 2259-2271.

Zhu, X., Oguh, A., Gingerich, M.A., Soleimanpour, S.A., Stoffers, D.A., and Gannon, M. (2021). Cell Cycle Regulation of the Pdx1 Transcription Factor in Developing Pancreas and Insulin-Producing beta-Cells. *Diabetes* *70*, 903-916.

REVIEWERS' COMMENTS

Reviewer #1 (Remarks to the Author):

The authors have been very responsive to the reviewers' comments. Notably, in response to my comments they did the right experiments addressing the likely heterogeneity of endocrine precursors in the NeuroD1 conditional knock-out. They performed a single-cell transcriptome comparison of the control cells and the NeuroD1 conditionally inactivated cells. The only concern is the use of a different Cre line (driven by *Isl1*) which likely reduces the phenotype since they live longer postnatally (hence this choice). However this critique is moderated by the strong molecular phenotype observed. On the positive side, this allows to explore a stage later as compared to the bulk sequencing. The findings are interesting and establish better the molecular state in which the cells that have initiated the endocrine precursor state (NeuroD1 transcript initiation) reside in the absence of NeuroD1. Among the interesting findings are the coexpression of some hormones, imbalance of hormonal subtypes and the fact that a subset of beta and alpha cells have an altered identity. Surprisingly this altered identity affects a set of genes but some maturation genes have a normal expression. There are some surprising findings such as the persistence of an Neurog3+ EP population in the wt. This is likely a mislabeling of the cluster. It does express high levels of somatostatin and this cluster is probably made of δ cells. It is interesting that they seem to co-express low levels of Neurog3 but the authors may have to check this carefully as it may be one or two transcripts. This is something important to revisit prior to publication in the figure and in the text. Molecularly they don't appear much affected by the absence of NeuroD. The single cell transcriptome is complemented by the new immunohistochemistry data showing that they also express Sox9. The revisions also improved the quantitative characterization of the endocrine differentiation failure in Fig. 1.

Moreover they addressed important comments from the other reviewers including:

- A better characterization of the Cre line and its possible phenotypes
- A better characterization of the cells targeted by the NeuroD1-Cre allele over time
- A broadening of context in the introduction with the inclusion of the role of NeuroD1 in neurons.

Some methodologies could be improved. For example in Figure 1f counting the total number of endocrine cells in "the" central section of the pancreas as P0 is not accurate and accordingly the numbers vary. It would have been better to count more sections per animals and less animals and apply a normalization factor (to surface of section for example). However, the meaning of/a.u. in this figure is not clear and this may be a normalization. Nevertheless, it clearly shows the reduction of α and β cells and can stay as such. The tunnel in Figure S3 is moderately convincing and a positive control would have helped.

In Figure 3a, the proportion of tomato cells expressing insulin or glucagon seems to increase. Were the two columns inverted?

Line 70: when refereeing to the expression of Neurog3, "transient" would be more appropriate than "limited"

Line 122: no "s" at in combination

Lines 356-357: One may want to be careful with the following statement: "We specifically focused on adult islets of Langerhans because adult b-cells are generally homogeneous at the transcriptomic level 62." as there are plenty of single-cell sequencing papers arguing for heterogeneity. Sequencing in the adult was however a good idea as asynchrony of generation may be buffered.

Line 443: "... a marked reduction..."

Taken together, the confirmation of previous observations using a different mouse line and an interesting characterization of the cells accumulating in a partially differentiated state makes this article strong. The only important point to correct (or better argue for if the author disagree with my statement) is the mislabeling of the EP cells in the single-cell analysis and need to relabel into SST and discuss accordingly in the text.

Reviewer #2 (Remarks to the Author):

The authors have addressed all of my concerns.

Reviewer #3 (Remarks to the Author):

The authors have addressed all my concerns. I have no further concerns.

Reviewer #4 (Remarks to the Author):

The Authors have largely addressed the concerns about clarity and data quantifications brought up in the initial review. The Introduction and Discussion sections now better provide context and significance of the results. The added single cell transcriptomics data elucidate the unclear cell state that the Authors alluded to in bulk transcriptomics data. It would be helpful to also see a validation of the single cell results with stainings of a handful of endocrine and non-endocrine markers that will also provide some spatial context.

Remaining comments:

In Figure 1, the Authors quantify changes in cell number from their images, but the metric used appears to be inconsistent. Line 128 refers to numbers of cells quantified, whereas in the figures often density or number of cells per defined region are reported. It should be made clear what the quantifications represent, number vs. density, and within what context numbers of cells are changing. In Figure 1i, it would be helpful to point out where the alpha-mantle is in the control that is missing in the Neurod1ST condition.

Line 148 refers to changes in morphology of the endocrine cell mass. The Authors could provide more context for what these changes in morphology mean for the physiology of the endocrine cell mass. E.g. how does being organized into "strings/sheets along the invisible lines of pancreatic ducts" affect the functioning of islets of Langerhans in the 3D environment of the pancreas?

In line 209, what is the significance of the analysis of RNAseq data finding effects on the KEGG pathway? Defining KEGG would help.

How prevalent is the co-expression of Sox9 and Pdx1 shown in Figure 4i? A quantification of the imaging data would help.

Figure references for the Reactome enrichment mentioned in line 216 are needed.

Figure 4g needs an axis title.

Line 486 'chromatic landscape' should read 'chromatin landscape.'

REVIEWER COMMENTS

Reviewer #1 (Remarks to the Author):

The authors have been very responsive to the reviewers' comments. Notably, in response to my comments they did the right experiments addressing the likely heterogeneity of endocrine precursors in the NeuroD1 conditional knock-out. They performed a single-cell transcriptome comparison of the control cells and the NeuroD1 conditionally inactivated cells. The only concern is the use of a different Cre line (driven by *Isl1*) which likely reduces the phenotype since they live longer postnatally (hence this choice). However this critique is moderated by the strong molecular phenotype observed. On the positive side, this allows to explore a stage later as compared to the bulk sequencing. The findings are interesting and establish better the molecular state in which the cells that have initiated the endocrine precursor state (NeuroD1 transcript initiation) reside in the absence of NeuroD1. Among the interesting findings are the coexpression of some hormones, imbalance of hormonal subtypes and the fact that a subset of beta and alpha cells have an altered identity. Surprisingly this altered identity affects a set of genes but some maturation genes have a normal expression. There are some surprising findings such as the persistence of an Neurog3+ EP population in the wt. This is likely a mislabeling of the cluster. It does express high levels of somatostatin and this cluster is probably made of δ cells. It is interesting that they seem to co-express low levels of Neurog3 but the authors may have to check this carefully as it may be one or two transcripts. This is something important to revisit prior to publication in the figure and in the text. Molecularly they don't appear much affected by the absence of NeuroD. The single cell transcriptome is complemented by the new immunohistochemistry data showing that they also express Sox9. The revisions also improved the quantitative characterization of the endocrine differentiation failure in Fig. 1.

Moreover they addressed important comments from the other reviewers including:

- A better characterization of the Cre line and its possible phenotypes
- A better characterization of the cells targeted by the NeuroD1-Cre allele over time
- A broadening of context in the introduction with the inclusion of the role of NeuroD1 in neurons.

We appreciate your positive feedback on our responsiveness to reviewers' comments and the efforts made to address the heterogeneity of endocrine precursors in the NeuroD1 conditional knock-out. We agree that the use of a different Cre line may have implications, but it was chosen to account for the longer postnatal survival, which can help in exploring a stage later compared to bulk sequencing.

1) Some methodologies could be improved. For example, in Figure 1f counting the total number of endocrine cells in "the" central section of the pancreas as P0 is not accurate and accordingly the numbers vary. It would have been better to count more sections per animals and less animals and apply a normalization factor (to surface of section for example). However, the meaning of/a.u. in this figure is not clear and this may be a normalization. Nevertheless, it clearly shows the reduction of α and β cells and can stay as such.

We added the explanation for the normalization and the meaning of a.u. to clarify this.

2) The tunnel in Figure S3 is moderately convincing and a positive control would have helped.

Based on your recommendation, we have now added the positive control for TUNEL labeling (Figure S3).

3) In Figure 3a, the proportion of tomato cells expressing insulin or glucagon seems to increase. Were the two columns inverted?

We apologize for any confusion caused and appreciate the opportunity to clarify this matter. In Figures 3a and 3b, we presented tdTomato positive cells, which do not express insulin or glucagon at E15.5. These data were included to demonstrate that the mutant pancreas has a significantly lower number of hormone-producing cells, indicating an altered or disrupted differentiation process of endocrine cells in the mutant condition.

To avoid any misunderstanding, we have now improved the labeling on the graph, making it clear that the percentage of tdTomato cells not producing insulin and glucagon was quantified. The revised labeling should provide a better understanding of the data and the specific cell population being analyzed.

4) Line 70: when refereeing to the expression of Neurog3, “transient” would be more appropriate than “limited”

Line 122: no “s” at in combination

Lines 356-357: One may want to be careful with the following statement: “We specifically focused on adult islets of Langerhans because adult b-cells are generally homogeneous at the transcriptomic level 62.” as there are plenty of single-cell sequencing papers arguing for heterogeneity. Sequencing in the adult was however a good idea as asynchrony of generation may be buffered.

Line 443: “... a marked reduction...”

Thank you for bringing these corrections and inaccuracies to our attention. The text has been revised accordingly.

5) Taken together, the confirmation of previous observations using a different mouse line and an interesting characterization of the cells accumulating in a partially differentiated state makes this article strong. The only important point to correct (or better argue for if the author disagree with my statement) is the mislabeling of the EP cells in the single-cell analysis and need to relabel into SST and discuss accordingly in the text.

Thank you for your positive evaluation of our manuscript. Regarding your important point about the mislabeling of the EP cells in the single-cell analysis, we completely agree with your observation that the cluster represents delta (δ) cells, primarily based on their high somatostatin expression. Additionally, we have included a new reference (Gribben et al., 2021) that reports co-expression of *Sst* and *Rbp4* (additional marker of δ cells), providing further support for our accurate classification of this cluster as δ cells. In agreement with our data, their scRNA-seq analyses of adult islets identified *Neurog3* expressing cells within the δ cell cluster. We are grateful for your keen observation, which has improved the accuracy and interpretation of our findings.

Reviewer #2 (Remarks to the Author):

The authors have addressed all of my concerns.

Reviewer #3 (Remarks to the Author):

The authors have addressed all my concerns. I have no further concerns.

Reviewer #4 (Remarks to the Author):

The Authors have largely addressed the concerns about clarity and data quantifications brought up in the initial review. The Introduction and Discussion sections now better provide context and significance of the results. The added single cell transcriptomics data elucidate the unclear cell state that the Authors alluded to in bulk transcriptomics data. It would be helpful to also see a validation of the single cell results with stainings of a handful of endocrine and non-endocrine markers that will also provide some spatial context.

Remaining comments:

1) In Figure 1, the Authors quantify changes in cell number from their images, but the metric used appears to be inconsistent. Line 128 refers to numbers of cells quantified, whereas in the figures often density or number of cells per defined region are reported. It should be made clear what the quantifications represent, number vs. density, and within what context numbers of cells are changing.

We apologize for the inconsistencies in the metric used for quantification and any confusion caused by the usage of "number" and "density" interchangeably. We checked the text and figure legend and corrected the description accordingly to the quantification of number of cells or density.

2) In Figure 1i, it would be helpful to point out where the alpha-mantle is in the control that is missing in the *NeuroDIST* condition.

We have included extra labeling in the figure to specifically point out the alpha-cell mantle in the control that is missing in the *NeuroDIST* condition. This additional labeling should aid in better understanding the observed differences between the two conditions.

3) Line 148 refers to changes in morphology of the endocrine cell mass. The Authors could provide more context for what these changes in morphology mean for the physiology of the endocrine cell mass. E.g. how does being organized into “strings/sheets along the invisible lines of pancreatic ducts” affect the functioning of islets of Langerhans in the 3D environment of the pancreas?

In response to the comment, we have provided more context on the changes in morphology of the endocrine cell mass and its implications for the physiology of the islets of Langerhans. We have elaborated on the abnormal distribution of the islets of Langerhans and its potential negative effects on endocrine cell function (lines 150-159 of the revised manuscript).

4) In line 209, what is the significance of the analysis of RNAseq data finding effects on the KEGG pathway? Defining KEGG would help.

We performed functional profiling of the differentially expressed genes identified in our bulk-RNA-seq. For this purpose, we utilized g:Profiler, a widely used public web server known for its capability to perform large-scale functional characterization of gene lists (the citation is the Method section). The g:Profiler tool leverages multiple data sources and databases, including:

- a) Gene Ontology (GO) - This provides information on the molecular function, cellular component, and biological processes associated with genes.
 - b) Biological Pathways (KEGG Pathways, Reactome, and WikiPathways) - KEGG, in particular, stands for the Kyoto Encyclopedia of Genes and Genomes, which is a comprehensive database that catalogs information on molecular pathways and networks. It allows us to gain insights into the intricate interactions between genes and their involvement in various biological processes.
 - c) Regulatory Motifs in DNA (TRANSFAC and miRTarBase) - This provides information about regulatory elements and microRNA targets that can influence gene expression.
- By analyzing our RNAseq data with g:Profiler, we were able to identify statistically enriched functional terms associated with the differentially expressed genes. These functional terms include GO categories, and biological pathways (such as KEGG pathways), along with their corresponding p-values.

To ensure clarity and help readers understand the significance of KEGG, we have included a reference to the KEGG pathway database in the text. This allows interested readers to explore and gain a more in-depth understanding of the specific biological pathways involved in our study.

Once again, we appreciate your feedback and have made the necessary revisions to enhance the manuscript's clarity.

5) How prevalent is the co-expression of Sox9 and Pdx1 shown in Figure 4i? A quantification of the imaging data would help.

We included the quantification of the co-expression of SOX9 and PDX1.

6) Figure references for the Reactome enrichment mentioned in line 216 are needed.

We have added figure reference in the text and modified the figure by including color-coded marks representing different functional information sources from g:Profiler analyses (Fig. 4c, d).

7) Figure 4g needs an axis title.

We added the axis title.

8) Line 486 'chromatic landscape' should read 'chromatin landscape.'

Thank you. We corrected it.